# The Role of Mutual Guarantee Institutions in the Financial Sustainability of New Family-Owned Small Businesses

**Concepción de la Fuente-Cabrero** , **Mónica de Castro-Pardo \*** , **Rosa Santero-Sánchez** and **Pilar Laguna-Sánchez**

Faculty of Law and Social Sciences, University Rey Juan Carlos, 28032 Madrid, Spain;
concepcion.delafuente@urjc.es (C.d.l.F.-C.); rosa.santero@urjc.es (R.S.-S.); pilar.laguna@urjc.es (P.L.-S.)
\* Correspondence: monica.decastro@urjc.es; Tel.: +34-91-488-88-20

**Abstract:** Small family-owned companies are the most common type of European business structure and are characterised by their orientation to long-term goals. Therefore, they can play an important role in the launching of businesses related to sustainable growth. However, access to finance is difficult for start-ups. Mutual Guarantee Institutions (MGIs) mitigate this problem by facilitating long-term guaranteed loans, but they must assume responsibility for default losses. This paper analyses, as of the end of 2018, the loan default of the portfolio of guarantees formalised by Spanish MGIs with new companies between 2003 and 2012, a period including both economic growth and recession. The objective is to identify the annual evolution and the average global cost of default, as well as the differences in said portfolios according to the purpose of the loan, company size and economic activity. The analysis was developed while considering two scenarios: one determinist, using a ratio method and another stochastic, using an analysis of variance. We found differences in the distribution of defaults for the variables company size and sector of activity. The findings provide relevant information for managers and Public Administrations to improve the distribution of guarantees between Spanish MGIs and public institutions, and their coverage of Small and Medium Enterprise (SME) loan defaults.

**Keywords:** family new businesses; financial sustainability; access to finance; Mutual Guarantee Institutions; loan default

## 1. Introduction

Sustainable finance in the European Union is needed to improve the contribution of finance towards sustainable, inclusive growth, which is Goal 8 of the 2030 Agenda for Sustainable Development. The EU wants the current financial system to be better aligned with its sustainable growth policies and also to protect the financial system from sustainability risks [1–3]. All the financial intermediaries (FIs) have a key role to play to reach this objective.

Sustainable finance is strongly linked to financial sustainability, which is related to structural, relational and cognitive aspects that create value for a company [4] and the long-time survival capacity of companies. The link between business sustainability and the financial performance of a firm has been demonstrated by several studies, as reference [5] showed.

Sustainability Development Goals (SDG) include development-oriented policies to foster the formalization of micro-, small- and medium-sized enterprises in SDG8. Accessing financial services and expanding access to banking are specifically included in these policies [6].

Lack of access to credit needed for SMEs to be able to survive, especially in the context of financial stress, is a serious problem. It is more difficult and costly for SMEs to access credit than it is for larger



companies, even in developed countries [7,8]. In this sense, one of the main problems faced by SMEs, especially young companies, is the lack of collateral, which hinders access to finance. This question is essential to ensure the sustainability of family run Small and Medium Enterprises (SMEs). On one hand, the availability of external funds is necessary for the creation and impulse of new businesses. On the other hand, credit is essential for the maintenance of these firms during their first years of life, under conditions of financial stress and to ensure intergenerational transition [9]. Likewise, insufficient financial literacy has been recognised as one of the main reasons for the failure of SMEs [10] and as a barrier to the sustainable development for these companies [7]. [11] identified the impact of the lack of finance on business organisations in the start-up period. The role of knowledge-based resources in promoting sustainability in SMEs is a very contemporary issue and financial literacy has been considered as the key to financial decision making [7].

Small Family Business Start-ups are particularly sensitive to access to credit. Although some studies have shown the performance of family firms during the global financial crisis was better than non-family firms [12,13], however, this only occurred with multi-generational firms. Newly created firms, i.e., first-generation family businesses performed worse during the crisis. These firms commit more resources and take higher risks than multi-generational enterprises [14,15].

The importance of considering a long-term temporal horizon in financial analysis, i.e., the financial sustainability of SMEs, has been supported by some studies [16]. Business growth can be considered sustainable if it increases the long-term, economic, social and environmental capital of a company. For this reason, ensuring the availability of external funds in these periods is particularly important for the sustainability of small family businesses.

The immense importance of Small Family Businesses in Europe is unquestionable [17–19]. In terms of relative importance, SMEs form the basis of the European economy. Family businesses make up more than 70% of all European enterprises and play an important role in the dynamism and strength of European economies, sustainability and long-term stability [20].

The link between SMEs and family businesses is evident and many of the challenges facing family firms are shared with SMEs. In Europe, family enterprises are dominated by micro enterprises and SMEs [21]. In Italy, 93% of manufacturing companies with less than 50 employees are family businesses. Family companies comprise 60% of all enterprises in Germany and make up 55% of the country's GDP [20]. In Spain the importance of family businesses is increasing as 95% of SMEs are family companies. These businesses contribute about 57% of GDP and generate about 67% of private employment [21].

To promote the creation and financial sustainability of these enterprises, it is essential to ensure access to long-term credit. Some studies found internal resources to be the best alternative for entrepreneurship [22,23]. However, in some European countries, such as Spain, entrepreneurs depend heavily on financial institutions to obtain external resources [24]. Financial credit formalised through banks has a very important weight in their financial structure and is necessary for start-ups [25–28].

Nevertheless, the lack of guarantees related to unpaid credit risk is usually an obstacle that prevents the granting of loans [29]. This strangulation is motivated by the strong multi-level regulation currently at work in the financial sector to define resources and credit policies [30,31]. The measures adopted regarding risk-based capital requirements in the Basel II and Basel III Accords created a new framework with strong banking regulations and effects on the supply of credit [32,33]. This context is especially complex for small family businesses as financial constraints are greater for SMEs than for large firms and this makes it difficult for them to enter and become consolidated in markets.

Both problems have led policy makers to focus on solutions that help alleviate the financial constraint on SMEs. Fortunately, the creation of Mutual Guarantee Institutions (MGIs) has helped to mitigate this problem, providing additional guarantees for loans and permitting the sharing of credit risk responsibilities. The World Bank (Washington, USA) defines MGIs as a collective of independent enterprises and organisations that provide collective guarantees [34]. Currently, European MGIs support about 3 million SMEs associated to the European Association of Guarantee Institutions [35]

and have provided a strong impulse to the development and the sustainability of SMEs in European regions [36]. Spanish MGIs, which are considered to be FIs (supervised by the Spanish central bank), provide guarantees for long-term loans and also provide advice and financial literacy for SMEs [37]. Some of them have created a department to develop this area, such as ELKARGI, the first in the activity ranking [38].

In Europe, MGIs have institutional guarantees derived from national governments. Specifically, in Spain, the *Compañía Española de Reafianzamiento* (CERSA) is the institution that assumes a percentage of the loan defaults granted by MGIs. Therefore, the total risk of the credit destined to be provided to small family businesses is shared between the MGIs and CERSA [39].

Moreover, some European MGIs share risks with regional institutions too, as in the case of Germany. In turn, CERSA shares the risk with The European Investment Fund (EIF, Luxembourg) which offers securitizations of SME debt finance portfolios with guarantee institutions. This allows CERSA to take more risks with Spanish MGIs, and these MGIs can provide more loans to the SMEs [40]. CERSA has had the EIF securitization of their annual portfolios since 2000 [41].

MGIs contribute to making access to credit for Small Family Business easier in two ways: (i) reducing the credit risk provision and default for the financial institution, and (ii) decreasing the bank capital requirements in the loans granted by the MGIs. Both reduce the banking costs related to loans, so financing for SMEs therefore experiences improvement [41–43].

Despite the importance of these institutions regarding the development and financial sustainability of SMEs, MGIs have not been studied in depth, especially the cost of loan defaults, a key factor for the sustainability of these FIs. To our knowledge, there is no recent research about this subject. This implies the existence of a large gap in financial information regarding the default behaviour supported by these institutional guarantees. In the context of MGIs, this supposes two problems: MGIs need to have enough resources available to respond to the responsibility assumed and cover the losses related to defaulting on the granted loan, but there is no information available regarding this, and (2) the lack of information regarding default makes it very difficult to promote development-oriented policies. The need for pinpoint financial knowledge regarding the financing of SMEs by MGIs in Spain was the main incentive for this research.

The objective of this study is to characterise loan defaults and the losses related to financing provided by Spanish MGIs to new small family businesses during 2003–2012, a long period including a cycle change with economic growth and recession and to address a gap in the literature about MGIs. To achieve this objective we developed two analyses: (i) a global analysis of the annual evolution and the average global cost of loan default using a determinist scenario and (ii) a group comparison analysis, with a classification of credit into three groups: the purpose of the loan, company size and economic activity, hence providing a stochastic approach.

This study contributes to the literature on access to finance and the default costs of new SMEs, useful to promote development-oriented policies regarding SDGs. Furthermore, the research expands knowledge about MGIs contribution to access to credit, advice and financial literacy to SMES. In addition, the findings provide relevant information for the financial sustainability of these FIs, which is valuable for managers and Public Administrations, including EIF.

## 2. Background and Hypothesis

The analysis of delinquency in companies has been widely studied from the perspective of financial institutions [44]. Most of these studies seek to identify the causes of non-payment [45,46] and some of these have also proposed models for making predictions [47,48]. The behaviour of family businesses in relation to non-payment has received a lot of attention [49–52] and in particular, the effect of variables such as the size of the company and the financial structure of the SME have been extensively studied [53]. References [54,55] developed different works on the behaviour of start-ups considering delinquency. The first ones demonstrated the relevance of soft variables such as management style

on delinquency. The latter ones analysed the attitudes of banks when granting credit to this type of company.

Most of these studies make comparisons between small and large companies in terms of defaults, however, the comparative effect of other variables such as the purpose of the loan, the size and type of the economic sector on defaults has not been studied in depth in family businesses. Some papers have analysed the effect of some of these variables separately. For example, reference [56] determined some factors that contributed to defaults by microenterprises in a region of Ethiopia.

Reference [57] concluded that larger companies show a higher index of self-finance. However, smaller companies display precarious financial behaviour, apparently based on short-term financing. Reference [58] found that the larger the company, the more likely it is that the company is diversified and its financial flows are less volatile. Nevertheless, reference [59] argued that asymmetric information problems are greater in SMEs, leading them to endure financial constraints, which in turn translates into a greater tendency to short-term financing and less in the long-term. These authors conclude that business size has an influence on short-term debt, but not on long-term debt. Reference [60] considered that larger enterprises present higher capitalization rates and, consequently, lower levels of borrowing. This disparity in the results of empirical work in terms of size reflects a marked ambiguity in the relationship between size and indebtedness in SMEs due to the complex relationships that link them.

The influence of the sector of activity has also been widely studied, but no conclusive results have been obtained in this area either. Reference [61] analysed the default behaviour of 19,628 Italian companies in the manufacturing sector and identified more delinquency in companies that produced transport equipment and radio, TV and communication equipment. This analysis considered companies of all sizes. Reference [62] also analysed the causes of delinquency using a database of 6482 SMEs in the hospitality sector in Portugal mainly considering financial ratios.

Reference [59] pointed out that sectors where assets are mostly intangible and risky tend to borrow little, as opposed to sectors where assets are tangible and relatively safe. Reference [60] concluded that there were no significant differences between groups, except in the construction sector; moreover, industrial enterprises had higher borrowing rates than service sector enterprises.

On the other hand, and despite its importance for small businesses, few studies have been carried out about this issue in European MGIs. Some authors have shown that SMEs with the additional guarantee of MGIs pay less for credit than non-insured companies [25]. Reference [36] analysed the impact of the financial constraints on SMEs considering UniCredit loans and they identified the very important support of MGIs beyond just guarantees. Reference [42] analysed some regional differences in the impact of guaranteed credit supported by MGIs in 6157 firms. Default credit behaviour in Small Family Business Start-ups with MGI guarantees have not been studied in depth; however, good knowledge about this problem could be key to improving access to credit for these companies to ensure financial sustainability and intergenerational transition.

Based on the revised background, two hypotheses are proposed in this study to identify the behaviour of defaults over time, in periods of expansion and recession, and whether the purpose of the loan, company size and economic activity have significant effects on the volume of defaults.

**Hypothesis 1.** *Default behaviour in SME credit guaranteed by MGIs changes in different ways in contexts of stability or financial stress.*

**Hypothesis 2.** *Default behaviour in SME credit guaranteed by MGIs is influenced significantly by the purpose of the loan, company size and business sector.*

The first hypothesis was validated using a descriptive approach. The second hypothesis was validated using a stochastic analysis.

From these hypotheses, it is possible to describe the evolution of default behaviour in relation to time in the considered period, and to conclude whether the purpose of the loan, company size and/or economic activity significantly influence the defaults of family companies.

## 3. Materials and Methods

### 3.1. Database

The database analysed in this study was made up of 24,837 loans granted to Start-up SMEs supported by MGIs and guaranteed by CERSA during the period 2003–2012. This period includes some years characterised by economic growth and stability (2003–2007) and some years with economic recession (2008–2012). CERSA defines start-ups as newly created companies with less than 3 years of activity on the date of formalization of the guaranteed loan. The period 2003–2012 was chosen to work with practically expired portfolios, considering that the loans have an average duration of approximately 6 years. It is also a sufficiently long period to include periods of financial stress and stability. The database contains cross-sectional data, where each line of information comprises a company's financial operation and includes the characteristics of the financing guaranteed by the MGI: amount, date of formalisation and maturity, purpose of the operation (investment or working capital) and amount of the default; together with the characteristics of the guaranteed company: years in business, number of employees and sector of activity. Usually, the formalised operations are associated with a single company, although there is the case of several loans from/to the same company, so the database is not of companies, but of loans guaranteed by SGR.

### 3.2. Variable Definitions

Three main variables were identified and grouped by the annual portfolio, assessed in units and monetary guarantees. Moreover, a secondary variable was defined to obtain the cost of defaults of each annual portfolio.

Main variables:

Annual production of guarantees in numbers (units). Guaranteed loans, credit lines and financial leasing are included.

Annual production of guarantees by volume (Euros).

Default: Delinquent guaranteed loans paid off by MGIs to credit institutions, totally provisioned through the Technical Provision Fund (TPF) and rated as defaults in the data base for the annual production of guarantees.

Secondary variables:

Total Default Ratio of the annual portfolio: percentage that defaulted in the annual guarantee portfolio, regardless of the year in which the non-payment occurred, the allocation of the provision and the rating as default. The terms default ratio and default cost are used indistinctly throughout this work.

In addition, the loans were classified into three categories: purpose of the loan, company size, and economic activity. This grouping was carried out on the basis of the information supplied by the database in relation to the characteristics of the guaranteed operations and the guaranteed companies. This classification was useful for both descriptive and stochastic scenarios.

Purpose of the loan: classified into two groups: investment capital and working capital.

Company size: sub-classified into three groups; micro companies (0–9 employees), small companies (10–49 employees) and medium size companies (50–250 employees). This classification was selected as some studies have found differences related to the behaviour of firms with 9 or less employees when compared to enterprises traditionally considered to be small companies [38].

Economic activity: sub-classified into 7 groups following the Statistical Classification of Economic Activities in the European Community, Rev. 2 (2008) classification; primary sector, manufacturing, construction, wholesale and retail trade, transport and communications, accommodation and food service activities, and other services.

*3.3. Methods*

SPSS v.25 software was used for the primary treatment of the data, descriptive analysis and stochastic treatment of the data, using the variables described in this section.

The descriptive analysis was carried out using group and yearly portfolio default ratios. These ratios were calculated using Equations (1) and (2), respectively. The Group Default Ratio (GDR) was calculated for each group using Equation (1).

$$GDR_i^n = \frac{D_i^n}{L_i^n} x100 \tag{1}$$

where $GDR_i^n$ is the Group Default Ratio of the guaranteed portfolio assessed in year $n$ included in group $i$, $D_i^n$ is the monetary volume of defaults in this portfolio assessed in year $n$ included in group $i$, and $L_i^n$ is the total of the monetary volume of loans granted in year $n$ included in group $i$.

In a similar way, the Default Ratio (DR) per year was calculated using a weighted sum detailed in Equation (2).

$$DR_n = \sum_{i=1}^{n} w_i GDR_i^n \tag{2}$$

where $DR_i^n$ is the Default Ratio in the guarantee portfolio assessed in year $n$, $w_i$ is the weight of group $i$ and $GDR_i^n$ is the Group Default Ratio in year $n$ included in group $i$

Analysis of financial ratios has traditionally been used in financial research and when the sample is large enough, it is sufficient to characterise investment portfolios [52,63,64] In this sense, reference [59] only used a descriptive analysis to identify the effect of moral hazard on the loan characterisation of Colombian SMEs using financial ratios.

A stochastic analysis considering inter-group comparisons was made using an analysis of variance (ANOVA). The distribution of the variables was considered to have a normal distribution, as the database size was large enough. ANOVA analysis is a technique equivalent to linear regression when explanatory variables are categorical, as is the case here, and explains how much of the variation of the default volume is explained by size, sector or purpose of the loan. This analysis has been widely used to identify differences between groups and is considered an indicator that complements descriptive analysis. Examples of studies that only used an ANOVA or MANOVA analysis to identify differences in small enterprises are those of [57,59]. Others used a combined descriptive analysis and an ANOVA, such as reference [65], who used a ratio analysis and an ANOVA to identify the effect of energy suppliers on the financial structure of companies in this sector.

In this work, an ANOVA analysis was used to identify intergroup differences. To test homoscedasticity, we used a Levene test and to solve the problem of heteroscedasticity, the analysis of variance was realised using a Brown-Forsythe statistic. This test uses a modified F-statistic to analyse significant effects and is commonly used to deal with heteroscedastic samples [59]. Moreover, to identify intergroup differences we used a T3 Dunnett test, which is commonly used to make parametric post hoc tests with heteroscedastic and unequal samples [66].

## 4. Results

During the period analysed, 24,837 transactions were formalised for a total of 2,395,624,515 Euros in guarantees granted.

In total, 98% of the guarantees were granted by credit institutions, while 2% were granted by Public Organisations, such as the Centre for Technological and Industrial Development and the Centre for Environmental and Technological Energy Research, among others. 84.7% were long-term loans, 12.6% were medium-term, and only 2.7% were short-term. The average term of expiration of the operations was 87 months.

The defaults amounted to 220,239 Euros, with an average of 8867.40 and standard deviation of 69,606.30 (Table 1). Most of them were granted investment capital loans for microenterprises in the wholesale and retail trade sector, as well as other service sectors. The descriptive analysis also indicates a greater disparity in the amounts for loans destined to finance working capital, those of small businesses and those formalised for construction activities (Table 1).

**Table 1. Distribution of** Default (€) by Characteristics: average and standard deviation.

| Characteristics | Number of Loans | Default Average | Default Standard Deviation |
|---|---|---|---|
| **Purpose of the Loan** | | | |
| Working capital | 3346 | 11,408.71 | 112,015.73 |
| Investment capital | 21,491 | 8471.71 | 60,375.33 |
| Total | 24,837 | 8867.38 | 69,606.31 |
| **Company Size** | | | |
| Micro | 23,263 | 7649.36 | 60,839.75 |
| Small | 1397 | 27,383.46 | 150,999.15 |
| Medium | 177 | 22,810.69 | 102,522.97 |
| Total | 24,837 | 8867.38 | 69,606.31 |
| **Economic Activity** | | | |
| Primary sector | 1409 | 2542.89 | 21,871.20 |
| Manufacturing | 2822 | 11,452.06 | 57,323.89 |
| Construction | 2057 | 13,365.88 | 99,747.95 |
| Wholesale and retail trade | 6451 | 5340.08 | 31,714.65 |
| Transport and communications | 1875 | 3097.87 | 25,586.04 |
| Accommodation and food service activities | 3927 | 14,187.09 | 62,427.61 |
| Other services | 6296 | 9668.80 | 102,813.49 |
| Total | 24,837 | 8867.38 | 69,606.31 |

Source: The authors, based on data from CERSA.

Table 2 presents the number and volume of guarantees per year, the volume of default and the Total Default Ratio (TDR) for each guaranteed yearly portfolio. As we can see, the global default cost ratio for the whole period was 9.2 in 2018 (at the end of the year).

**Table 2.** Number and volume of guarantees per year, volume of default and Total Default Ratio (TDR) per guaranteed yearly portfolio.

| Year | Number | Guarantee (€) | Default (€) | TDR |
|---|---|---|---|---|
| 2003 | 2289 | 181,443,910 | 5,966,069 | 3.3 |
| 2004 | 3010 | 231,565,355 | 8,990,038 | 3.9 |
| 2005 | 2806 | 246,416,137 | 19,684,483 | 7.9 |
| 2006 | 3051 | 347,455,009 | 36,303,107 | 10.4 |
| 2007 | 2.642 | 346,740,155 | 44,014,557 | 12.7 |
| 2008 | 2475 | 312,607,985 | 47,531,806 | 15.2 |
| 2009 | 1977 | 205,352,586 | 21,535,737 | 10.5 |
| 2010 | 2.585 | 221,121,400 | 16,171,410 | 7.3 |
| 2011 | 2352 | 175,275,127 | 13,540,274 | 7.7 |
| 2012 | 1650 | 127,646,850 | 6501,665 | 5.1 |
| Total | 24,837 | 2,395,624,515 | 220,239,147 | 9.2 |

Source: The authors, based on data from CERSA.

During the first 4 years, the granted new guarantees followed an upward trend, specifically the annual portfolio in 2006 (€ 347,455,009) was 91% higher than in 2003 (€ 231,565,355). However, in 2007

there was stagnation and a subsequent reduction that coincided with the economic recession. In line with these results, the ratio for the portfolios formalised in 2003 and 2004 was relatively low at 3.3 and 3.9, respectively. However, from 2005 it began to rise considerably as a result of the economic crisis that occurred in 2008, reaching levels above 8 and attaining its highest level with the 2008 portfolio at 15.2. From then on, it began to descend gradually to 5.1 in 2012, although it still did not approach the values from the beginning of the period (Table 2).

Group Default Ratios regarding the purpose of the loan, company size and economic activity are shown in Table 3.

**Table 3.** Group Default Ratio (GDR) and Default Ratio (DR) per guaranteed yearly portfolio.

| Characteristics | | 2003 | 2004 | 2005 | 2006 | 2007 | 2008 | 2009 | 2010 | 2011 | 2012 |
|---|---|---|---|---|---|---|---|---|---|---|---|
| Purpose of the Loan | Investment Capital | 3.5 | 3.9 | 7.5 | 10.9 | 12.6 | 14.9 | 5.7 | 6.6 | 7.6 | 4.5 |
| | Working Capital | 1.3 | 4.3 | 12.2 | 7.7 | 13.5 | 18.1 | 25.6 | 11.5 | 8.4 | 9.0 |
| Company Size | Micro | 2.9 | 3.9 | 7.9 | 9.0 | 12.5 | 14.8 | 10.9 | 7.1 | 5.5 | 5.1 |
| | Small | 6.5 | 4.5 | 7.1 | 18.0 | 14.2 | 19.5 | 6.9 | 8.8 | 22.4 | 4.4 |
| | Medium | 0.0 | 1.3 | 14.2 | 14.9 | 9.1 | 7.1 | 18.0 | 6.3 | 11.3 | 7.7 |
| Economic Activity | Primary Sector | 0.8 | 2.6 | 2.9 | 5.7 | −0.4 | 7.7 | 0.1 | 6.9 | 2.3 | 0.6 |
| | Manufacturing | 4.9 | 5.3 | 7.7 | 10.7 | 9.4 | 9.4 | 7.4 | 8.6 | 7.1 | 9.6 |
| | Construction | 1.6 | 3.1 | 10.0 | 8.3 | 15.3 | 30.0 | 33.6 | 28.3 | 24.0 | 12.8 |
| | Wholesale and Retail Trade | 1.8 | 2.2 | 7.6 | 7.1 | 13.4 | 13.6 | 7.7 | 5.2 | 8.7 | 7.3 |
| | Transport and Communications | 11.5 | 5.3 | 1.6 | 5.8 | 5.2 | 9.3 | 0.9 | 1.0 | 1.0 | 2.4 |
| | Accommodation and Food Service Activities | 3.8 | 3.8 | 12.8 | 16.2 | 17.1 | 18.8 | 8.1 | 11.6 | 6.7 | 7.7 |
| Other Services | | 2.0 | 4.8 | 5.1 | 11.2 | 13.3 | 18.2 | 15.9 | 5.3 | 12.2 | 2.1 |
| Default Ratio (DR) | | 3.3 | 3.9 | 7.9 | 10.4 | 12.7 | 15.2 | 10.5 | 7.3 | 7.7 | 5.1 |

Source: The authors, based on data from CERSA.

Regarding the purpose of the loan, the GDR was higher for the guaranteed loans for working capital for almost all the years in the period except 2003 and 2006. The difference was more important in 2009, when the ratio of defaults was 25.6 compared to only 5.6 of the default credit for investment capital. The average defaults for the period were 7.8 for capital investment loans and 11.2 for working capital loans (Table 3).

Considering company size, small enterprises presented the highest default ratios, except in 2005, 2009 and 2012. In fact, the average default of small companies was 11.2, followed by medium-sized companies with 9. Microenterprises, with an average ratio of 8, had the lowest cost of credit risk (Table 3).

Finally, the economic activity group analysis shows that the default costs of the primary sector were very low, with an average GDR of 2.9 for the entire period, followed by transport and communications. Wholesale and retail trade (7.4), manufacturing (8), other services (9) and accommodation and food service activities (10.7) had higher average costs which were closer to the global cost. Finally, the highest cost of defaults was that of the construction sector, with some very high annual levels and an average ratio of 16.7 (Table 3).

To validate the robustness of the descriptive analysis a stochastic analysis was carried out. The analysis of variance permitted the identification of significant differences between company size and the economic activity groups throughout the period ($p < 0.05$). Moreover, post hoc testing provided more in-depth information about inter-group differences.

Three ANOVA analyses were applied, combining each factor with each variable to identify the significant effects of the purpose of the loan, company size and business sector on the default volume.

From these results, it was possible to conclude that company size and economic activity significantly influence the defaults of smaller companies as the null hypothesis was rejected for the second and third tests at a level of significance of less than five per cent (Table 4).

**Table 4.** Results of ANOVA analysis by group.

| Characteristics | Brown-Forsythe | Gl1 | Gl2 | Significance |
|---|---|---|---|---|
| Purpose of the Loan | 2.20 | 1 | 3653.27 | 0.138 |
| Company Size | 17.02 * | 2 | 1090.08 | 0.000 |
| Economic Activity | 14.98 * | 6 | 11,058.02 | 0.000 |

* Statistically significant at 99%. Source: The authors, based on data from CERSA.

After that, we carried out a post hoc analysis using the T3-Dunnett statistic for company size and economic activity groups to identify paired mean differences between groups.

Results related to the company size group showed differences between micro firms and the other two subgroups, with a lower risk for micro firms vs. small and medium firms (Table 5).

**Table 5.** T3-Dunnett results to identify intergroup differences regarding company size.

| Groups | Company Size | Mean differences | Standard Error | Significance |
|---|---|---|---|---|
| Micro firm | Small firm | −19,734.10 * | 4059.60 | 0.000 |
| | Medium firm | −15,161.33 | 7716.41 | 0.145 |
| Small firm | Micro firm | 19,734,10 * | 4059.60 | 0.000 |
| | Medium firm | 4572.77 | 8700.87 | 0.936 |
| Medium firm | Micro firm | 15,161.33 | 7716.41 | 0.145 |
| | Small firm | −4572.77 | 8700.87 | 0.936 |

* Statistically significant at 99%. Source: The authors, based on data from CERSA.

Finally, the specific analysis by economic activity showed significant differences between the primary sector and all the other subgroups except transport and communications, manufacturing and all the other subgroups except wholesale and retail trade and other services, and construction and all the other subgroups except wholesale and retail trade and accommodation and food service activities. The highest number of defaults was found in manufacturing, construction activities and accommodation and food service activities. The greatest differences were identified relating to primary and transport and communications activities (Table 6).

**Table 6.** T3-Dunnett results to identify intergroup differences considering economic activity.

| Economic Activity | Groups | Mean Differences | Standard Error | Sig. |
|---|---|---|---|---|
| Primary | Manufacturing | −8909.16 | 1226.35 | 0.000 |
| | Construction | −10,822.99 | 2275.19 | 0.000 |
| | Wholesale and retail trade | −2797.20 | 703.86 | 0.002 |
| | Accommodation and food service activities | −11,644.20 | 1154.08 | 0.000 |
| | Other services | −7125.91 | 1420.72 | 0.001 |
| Manufacturing | Primary | 8909.17 | 1226.35 | 0.000 |
| | Wholesale and retail trade | 6111.97 | 1149.06 | 0.000 |
| | Transport and communications | 8354.19 | 1230.28 | 0.000 |
| Construction | Primary | 10,822.99 | 2275.19 | 0.000 |
| | Wholesale and retail trade | 8025.79 | 2234.48 | 0.007 |
| | Transport and communications | 10,268.02 | 2277.30 | 0.000 |

**Table 6.** *Cont.*

| Economic Activity | Groups | Mean Differences | Standard Error | Sig. |
|---|---|---|---|---|
| Wholesale and Retail Trade | Primary | 2797.20 | 703.86 | 0.002 |
| | Manufacturing | −6111.97 | 1149.06 | 0.000 |
| | Construction | −8025.79 | 2234.48 | 0.007 |
| | Transport and communications | 2242.22 * | 710.68 | 0.033 |
| | Accommodation and food service activities | −8847.00 | 1071.60 | 0.000 |
| | Other services | −4328.71 * | 1354.57 | 0.030 |
| Transport and Communications | Manufacturing | −8354.19 | 1230.28 | 0.000 |
| | Construction | −10,268.02 | 2277.30 | 0.000 |
| | Wholesale and retail trade | −2242.22 * | 710.68 | 0.033 |
| | Accommodation and food service activities | −11,089.23 | 1158.26 | 0.000 |
| | Other services | −6570.93 | 1424.11 | 0.001 |
| Accommodation and Food Service Activities | Primary | 11,644.20 | 1154.08 | 0.000 |
| | Wholesale and retail trade | 8847.00 | 1071.60 | 0.000 |
| | Transport and communications | 11,089.23 | 1158.26 | 0.000 |
| Other services | Primary | 7125.91 | 1420.72 | 0.001 |
| | Wholesale and retail trade | 4328.71 * | 1354.57 | 0.030 |
| | Transport and communications | 6570.93 | 1424.11 | 0.001 |

* Statistically significant at 95%. Source: The authors, based on data from CERSA.

## 5. Discussion

### 5.1. Global Analysis of Loan Default

The annual analysis of the DRs permitted identification of the evolution of defaults by year and loan type for the whole period considered. Some studies have already demonstrated a significant relationship between loan default and some variables related to the economic context, such as the growth of the unemployment rate that increases the non-performing loans [63]. For this reason, the portfolio analysed is relevant, since it includes a broad period of 10 years, which includes at least one phase of economic growth and stability, and a phase of economic recession.

Likewise, the DR for the whole period 2003–2012 was 9.2 at the end of 2018. This global data on defaults is very useful to estimate their coverage in the funding of new family businesses and establish the necessary contribution of the Technical Provision Fund (TPF), whose purpose is to cover default in Spanish MGIs. The fees charged by banks to newly created small businesses are higher than to larger consolidated companies, to cover the risk of delinquency [67]. The information provided could allow financial institutions to better estimate the individual risk of each company and the TPF and, therefore, adjust commissions without penalising some companies exclusively due to their size. Reference [68] emphasises the need to provide financing alternatives to small businesses and propose a model to estimate a specific rating for this type of company. The information provided by the ratio analysis proposed in this study is in line with this strategy, since its added value is based on its ability to provide useful information to better define the risk profile of family businesses. With this information, policies can be articulated to finance the TPF used for the coverage of loan default with contributions in five directions: (a) contributions to the TPF by the MGIs for their coverage of part of the loan default; (b) contributions to the CERSA TPF for the part covered by this company; (c) counter-guarantees from the Autonomous Administrations that in some cases cover a percentage of the guarantee portfolio for the part not covered by CERSA; (d) contributions to specific TPFs by some municipalities or other public or private institutions that may have an interest in promoting support for certain entrepreneurial

initiatives, innovative, sectorial, youth, social economy or new businesses run by women [69], and finally (e) the percentage of the EIF securitization to CERSA in the annual portfolio of this company.

*5.2. Group Comparison Analysis*

Inter-group analysis permitted the identification of significant differences related to company size and economy activities.

The results regarding the purpose of the loan showed that the default rate for the period for loans destined for capital investment was 7.8, less than the 11.2 for working capital loans. In general, the GDR for these last operations were higher for almost every year in the period, being especially high for the 2009 portfolio, with a GDR of working capital operations of 25.6 compared to 5.6 of those destined to finance investment capital. This information is interesting and should be taken into account by MGI managers for two reasons. On the one hand, medium-term operations aimed at financing working capital are necessary for small businesses, especially for newly created ones. Although the analysis of variance did not detect significant differences between defaults for both types of loans, the dispersion of risk behaviour is greater for loans destined to finance working capital, and therefore it would be advisable to be prudent in decision-making processes and perform an in-depth specific risk analysis when formalising these operations; or even charge higher endorsement commissions in these cases.

Regarding non-payment by companies according to their size, some works have found significant differences between companies with different volumes of profits [70]. The results of this study corroborate these differences for companies of different sizes, while considering the number of employees. Specifically, there is better performance in microenterprises compared to other companies. Again, this information is interesting as it suggests lower endorsement commissions for this type of company.

Finally, the differences between activity sectors show a greater risk in operations related to construction, and accommodation and food service activities, while the costs of default are much lower in the primary sector and transport and communications. The strategic importance of the primary sector, especially in developing economies, has been supported by several studies [71]. However, access to rural loans is sometimes extremely difficult for family businesses, with consequences for output production, household income, and the reduction of poverty. The results show the good behaviour associated with non-payments in the primary sector. Based on this information, it would be advisable to apply different treatment, with lower guarantees and commission requirements, to the primary sector with respect to other sectors, such as construction and accommodation and food service activities, which have worse behaviour in terms of default.

Accommodation and food service activities is, together with the construction sector, the sector with the highest default rate, showing significant differences to the rest of the economic sectors. These activities are especially relevant in economies with important tourist activity. However, some studies have already demonstrated the bad behaviour of this sector regarding delinquency [57,72]. On the other hand, the high dependence of the construction sector on macroeconomic cycles, as well as the large number of loans, makes this a volatile sector with high risk.

These results are relevant for two reasons. Firstly, managers could consider applying different commissions to different sectors of activity depending on the cost of the default. Secondly, the need to make different contributions to the TPF could be considered according to the sector of activity, since they have a different default cost. In this case, the information is relevant, both for the managers and for the Public Administration that makes contributions to the TPF.

## 6. Conclusions

The importance of MGIs to ensure the financial sustainability of family businesses requires a good characterisation of the default credit risk for their successful operation and the ability to manage losses related to defaulted loans.

Between 2003 and 2012, Spanish MGIs assumed a cost of around 22 million Euros/year that implied a default ratio of 9.2. The majority of the defaulting enterprises were small businesses with 10–49 employees in companies related to accommodation and food service activities, construction, and the wholesale and retail trade.

Moreover, the use of stochastic analysis permitted the identification of significantly high default levels in companies related to construction, manufacturing, and accommodation and food service activities, compared to defaults in other economic sectors. The primary sector showed the best performance regarding defaults. The ANOVA carried out takes into account the entire period globally: crisis and expansion, but there could be different behaviour in each year that should be analysed in future research.

This study is a first approach for the estimation of the total default cost and for classification according to the purpose of the loan, company size and economic activity of the company. However, although the results were robust, this paper only focuses on Spanish firms and this limits the generalisation of our conclusions to some extent. It would be interesting to develop future transnational studies that define risk distribution, and to obtain probabilities that could help to pinpoint the estimation of the amount that should be assigned to the Technical Provision Fund.

This information could be interesting for the managers of MGIs and Public Administrations and improve risk management in these institutions so that they can continue supporting the financing of long-term family businesses. Thus, these results could help with the attainment of some GDS while contributing to improving access to credit for SMEs in terms of quality and cost, promoting development-oriented policies that support productive activities, decent job creation and entrepreneurship, strengthening the capacity of domestic financial institutions to encourage and expand access to banking, insurance and financial services for all and improving access to financial services, including affordable credit, for small-scale industrial and other enterprises.

**Author Contributions:** The authors have contributed equally to this work

**Funding:** This research received no external funding. The APC was funded by University Rey Juan Carlos

**Acknowledgments:** The authors of this study would like to thank CERSA for ceding the data for the development of this work, as well as its disposition and selfless collaboration. The authors are greatful for useful comments and suggestions from the editor and reviewer.

**Conflicts of Interest:** The authors declare no conflict of interest.

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
