# Peer review of "The Role of Mutual Guarantee Institutions in the Financial Sustainability of New Family-Owned Small Businesses"

_sustainability, doi:10.3390/su11226409_

Round 1
Reviewer 1 Report
Dear authors,
Although, this research could be an interesting contribution to the current knowledge regarding the SMEs (and New Family-Owned Small Businesses), there are some major concerns regarding the following aspects:
Financial sustainability and sustainable finance seem to be used interchangeably in this research? It is not clear at all how the ”sustainable finance” (line 14) paradigm is reflected within this research. Also, there is not emphasized the connection between the theme of the research and the Aims and Scope of the Sustainability journal (specifically, the main analysis does not highlight any sustainability issue as reflected within the Aims and Scope of the journal). More specifically, how does this analysis is of interest for the wide readership of Sustainability Journal? 1. Global analysis of loan default (line 298-324) seems to be merely a descriptive approach (and not developed with more complex technical standards, as generally required for mainstream researches). Is there any econometric background that could support its conclusions? This could inflict upon the criteria of having analyses performed with the highest technical standards. Further, are there any hypothesis formulated regarding this analysis? The originality and novelty aspects are poorly mentioned, without clearly mentioning the importance for the sustainability More specifically, there is not properly emphasized the advance in current knowledge related to sustainability and sustainable development. Why was the ANOVA employed for this analysis? There is no scientific justification for your methodological choice with academic examples (e.g. what are the advantages and limits of this method?). Was ANOVA used in similar studies (not necessarily for SMEs)? Using more complex techniques could improve the results? Why was the sample chosen only for the 2003-2012 period? In terms of language and scientific presentation, there must be improvements. For example, what does “in a specific way” explains in lines 300-301?Author Response
Report for Review 1
Dear authors,
Although, this research could be an interesting contribution to the current knowledge regarding the SMEs (and New Family-Owned Small Businesses), there are some major concerns regarding the following aspects:
First of all, thank you very much for your valuable comments. All your suggestions are very important, which have important significance for our writing and scientific research work. When revising the article, we have considered carefully what you said.
Financial sustainability and sustainable finance seem to be used interchangeably in this research? It is not clear at all how the ”sustainable finance” (line 14) paradigm is reflected within this research. Also, there is not emphasized the connection between the theme of the research and the Aims and Scope of the Sustainability journal (specifically, the main analysis does not highlight any sustainability issue as reflected within the Aims and Scope of the journal). More specifically, how does this analysis is of interest for the wide readership of Sustainability Journal?
We would like to thank you for your valuable contributions to improve this manuscript. As you have suggested improving the quality of the manuscript, several new paragraphs have been included in section 1 - Introduction to explain in more detail linking the concept of sustainability finance in the European Union context with the manuscript, including the support to MGIs Systems through the European Investment Fund. Furthermore, the interest of this study for the readers of Sustainability Journal has been emphasized and both the terms sustainable finance and financial sustainability have been clarified. (Lines 30-66, 114-117, 122-132, 134-136).
“Sustainable finance in the European Union is imperative to improve the contribution of finance for sustainable, inclusive growth (SDG8). The financial system has a key role to play and the EU wants the current financial system to be better aligned with its sustainable growth policies and to protect the financial system from sustainability risks [1-3].
Broadly speaking, the sustainability framework is based on environmental, economic and social issues related to present and future generations, including sustainable finance to make economic prosperity long-lasting [4-8]. Sustainable finance is strongly linked to financial sustainability, which is related to structural, relational and cognitive aspects that create value for a company [9] and the long-time survival capacity of companies. The difference between these issues is a qualitative shift that lies in implementing the sustainability of financial activities, including promoting social and environmental responsibility [10-11].
The Goal Development Sustainability (GDS) framework provides an umbrella that encompasses both ideas. Considering GDS, most social innovations case studies have been related to an improvement in health and well-being [12]. However, the importance of small and medium enterprises (SMEs) for economic development through wealth distribution, creation of employment, technological advancement, reduction of poverty and innovation [13] has contributed to the consideration of the development and sustainability in the GDS framework.
Specifically, these goals are: 8.3. to promote development-oriented policies that support productive activities, decent job creation, entrepreneurship, creativity and innovation, and encourage the formalization and growth of micro-, small- and medium-sized enterprises, including access to financial services, 8.10; to strengthen the capacity of domestic financial institutions to order to encourage and expand access to banking, insurance and financial services for all, and 9.3. to increase access for small-scale industrial and other enterprises, especially in developing countries, to financial services, including affordable credit, and their integration into value chains and markets [14].
These specific goals are to promote the development of SMEs and ensure their access to credit so that they can be sustained in the long term. In this regard, the approach of social sustainability and intergenerational transition in family businesses has been a much studied topic in the last ten years [15-17]. However, the succession of control in family businesses is especially complex [18-19], as they comprise multi-dimensional scenarios that include financial issues. Access to credit so that SMEs can survive, especially in financial stress contexts, is a serious problem, even in developed countries [20]. In this sense, one of the main problems faced by SMEs, especially young companies, is the lack of collateral, which hinders access to finance. This question is essential to ensure the sustainability of family run Small and Medium Enterprises (SMEs). On one hand, the availability of external funds is necessary for the creation and impulse of new businesses. On the other hand, credit is essential for the maintenance of these firms during their first years of life or under conditions of financial stress and to ensure intergenerational transition in these businesses [21].”
“In turn, CERSA shares the risk with The European Investment Fund (EIF) which offers securitizations of SME debt finance portfolios with guarantee institutions, this allows CERSA to take more risks with Spanish MGIs, and these MGIs can provide more loans to the SMEs [47]. CERSA has the EIF securitization of their annul portfolios since 2000 [48]”
“Despite the importance of these institutions regarding the development and financial sustainability of SMEs, MGIs have not been studied in depth. This implies the existence of a large gap in financial information regarding the default behaviour supported by these institutional guarantees. The role of the knowledge-based resources in promoting sustainability in SMEs is a very contemporary issue and financial literacy has been considered key for financial decision making [13]. In the context of MGIs, this supposes two problems: MGIs need to have enough resources available to respond to the responsibility assumed and cover the losses related to defaulting on the granted loan, but there is no information available regarding this, and (2) the lack of information regarding default makes it very difficult to promote development-oriented policies. The need for pinpoint financial knowledge regarding the financing of SMEs by MGIs in Spain was the main incentive for this research.”
“during 2003-2012, a long period including a cycle change with economic growth and recession, to generate financial literacy useful to promote development-oriented policies regarding SMEs.”
Furthermore, a little change on the research goal has done to clarify it in the Abstract (Line 17-19) and at the end of the Introduction (Lines 134-136).
“This paper analyses at the end of 2018 the loan default of the portfolio of guarantees formalized for Spanish MGIs with new companies in 2003-2012, a period including economic growth end recession.”
“during 2003-2012, a long period including a cycle change with economic growth and recession, to generate financial literacy useful to promote development-oriented policies regarding SMEs.”
Global analysis of loan default (line 298-324) seems to be merely a descriptive approach (and not developed with more complex technical standards, as generally required for mainstream researches). Is there any econometric background that could support its conclusions? This could inflict upon the criteria of having analyses performed with the highest technical standards. Further, are there any hypothesis formulated regarding this analysis?
Thank you for reading this article carefully and making valuable suggestions. As we said before we have include that CERSA has the EIF securitization of their portfolios then we have considered that it is necessary and important to expose as well, in 5.1 Global analysis of loan default, the utility of the research findings for de EIF and we have included this issue (Lines 395-396).
“and finally e) percentage of the EIF securitization to CERSA in the annual portfolio to this company”.
As you have said, the ratio analysis is a descriptive analysis. Although, considering the reviewed literature, the analysis carried out to develop this research is suitable and complete to characterize the failed loans in companies guaranteed by MGS in Spain. Data base is the population of all the new companies guaranteed by MGI in Spain. As the data represent the population, it is not necessary to develop more complex techniques to obtain reliable results. We have explained more about the database to clarify this question (Lines 202-213).
“CERSA defines start-ups as newly created companies with less than 3 years of activity on the date of formalization of the guaranteed loan. The period 2003-2012 was chosen to work with practically expired portfolios, considering that the loans have an average duration of approximately 6 years. It is also a sufficiently long period to include periods of financial stress and stability. The database contains cross-sectional data, where each line of information comprises a company’s financial operation and includes the characteristics of the financing guaranteed by the MGI: amount, date of formalisation and maturity, purpose of the operation (investment or working capital) and amount of the default; and the characteristics of the guaranteed company: years in business, number of employees and sector of activity. Usually, the formalised operations are associated with a single company, although there is the case of several loans from/to the same company, so the database is not of companies, but of loans guaranteed by SGR.”
On the other hand, until now, loan default behaviour in this type of company has not been studied, so it’s completely unknown. We provide a determinist approach using a ratio analysis and an ANOVA analysis, which provides a stochastic approach. Results from the ANOVA have a significance of 95% and 99% (Tables 2-5). We have provided some references regarding studies that used only ratio analysis or ANOVA analysis, or ratios and ANOVA techniques, in similar research (lines 263-266, 273-276). We are so sorry that we did not explain the use of ANOVA clearly where the analysis was applied in the original manuscript. We would really appreciate it if you would forgive our carelessness.
“Analysis of financial ratios has traditionally been used in financial research and when the sample is large enough, it is sufficient to characterize investment portfolios [70, 71, 59] In this sense, [59] only used a descriptive analysis to identify the effect of moral hazard on the loan characterization of Colombian SMEs using financial ratios.”
“This analysis has been widely used to identify differences between groups and companies and is considered an indicator that complements descriptive analysis. Examples of studies that only used an ANOVA or MANOVA analysis to identify differences in small enterprises are those of [64,66]. Some more recent works used a combined descriptive analysis and an ANOVA, such as [72], which used a ratio analysis and an ANOVA to identify the effect of energy suppliers on the financial structure of companies in this sector.
In this work, an ANOVA analysis was used to identify intergroup differences. To test homoscedasticity we used a Levene test and to solve the problem of heteroscedasticity, the analysis of variance was realised using a Brown-Forsythe statistic.”
Moreover, we have changed the hypotheses proposed in the original manuscript to two new hypotheses in order to clarify the logic of the research and to improve the quality of the manuscript, following the recommendation of the reviewers. We have added a first hypothesis solved with the ratio analysis and a second hypothesis solved with the analysis of variance (Lines 185-196).
“Based on the revised background, two hypotheses are proposed in this study to identify the behaviour of defaults over time, in periods of expansion and recession, and whether the purpose of the loan, company size and economic activity, have significant effects on the volume of defaults.
H1: Default behaviour in SME credits guaranteed by MGIs changes in a different way in contexts of stability or financial stress.
H2: Default behaviour in SME credits guaranteed by MGIs is influenced significantly by the purpose of the loan, company size and business sector.
The first hypothesis was validated using a descriptive approach. The second hypothesis was validated using a stochastic analysis.
From these hypotheses, it is possible to describe the evolution of default behaviour in relation to time in the considered period, and to conclude whether the purpose of the loan, company size and/or economic activity significantly influence the defaults of family companies.”
The originality and novelty aspects are poorly mentioned, without clearly mentioning the importance for the sustainability. More specifically, there is not properly emphasized the advance in current knowledge related to sustainability and sustainable development.
Thank you for reading this article carefully and making valuable suggestions, which have greatly improved and perfected our manuscript. The originality and novelty aspects and the importance for sustainability have been remarked on in the Introduction Section (Lines 122-132).
“Despite the importance of these institutions regarding the development and financial sustainability of SMEs, MGIs have not been studied in depth. This implies the existence of a large gap in financial information regarding the default behaviour supported by these institutional guarantees. The role of the knowledge-based resources in promoting sustainability in SMEs is a very contemporary issue and financial literacy has been considered key for financial decision making [13]. In the context of MGIs, this supposes two problems: MGIs need to have enough resources available to respond to the responsibility assumed and cover the losses related to defaulting on the granted loan, but there is no information available regarding this, and (2) the lack of information regarding default makes it very difficult to promote development-oriented policies. The need for pinpoint financial knowledge regarding the financing of SMEs by MGIs in Spain was the main incentive for this research.”
The advance in current knowledge related to sustainability has been noted in the Conclusions (Lines 463-468).
“Thus, these results could help with the attainment of some GDS while contributing to improve access to credit for SMEs in terms of quality and cost, promoting development-oriented policies that support productive activities, decent job creation and entrepreneurship, strengthen the capacity of domestic financial institutions to encourage and expand access to banking, insurance and financial services for all and improve access to financial services, including affordable credit, for small-scale industrial and other enterprises.”
Why was the ANOVA employed for this analysis? There is no scientific justification for your methodological choice with academic examples (e.g. what are the advantages and limits of this method?). Was ANOVA used in similar studies (not necessarily for SMEs)? Using more complex techniques could improve the results?
Thank you for your careful reading and valuable comments. The ANOVA was used in the analysis to complete the descriptive analysis. The ANOVA is performed for detecting the importance of the size, industry and propose of loan in fail ratio variation. This analysis explains how much of the variation of the variable of interest is explained by named variables. This method is useful in two ways, when the independent variables are categorical, and the inter-group differences are identified and hypothesis can be verified (Lines 269-273). Therefore, the authors consider that the methodology used is appropriate for the research objectives.
“ANOVA analysis is a technique equivalent to linear regression when explanatory variables are categorical, as is the case here, and explains how much of the variation of the default volume is explained by size, sector or propose of loan. This analysis has been widely used to identify differences between groups and is considered an indicator that complements descriptive analysis.”
We have added some references that only used an ANOVA or MANCOVA analysis to develop similar studies (Lines 273-276).
“Examples of studies that only used an ANOVA or MANOVA analysis to identify differences in small enterprises are those of [64, 66]. Others used a combined descriptive analysis and an ANOVA, such as [72], which used a ratio analysis and an ANOVA to identify the effect of energy suppliers on the financial structure of companies in this sector. “
Why was the sample chosen only for the 2003-2012 period? In terms of language and scientific presentation, there must be improvements. For example, what does “in a specific way” explains in lines 300-301?
Thank you very much for your advice. Following your valuable recommendations, the justification of the considered period has been explained in order to improve the scientific presentation. We chose 2003-2012 for two reasons, it is long period including a cycle change with economic growth and recession and it was important to work with practically expired portfolios, considering that the loans have an average duration of approximately 6 years, so the information provided by CERSA is the situation of the annual portfolios at the end of 2018. We have justified better this in Abstract (Line 17-19), Introduction (Lines 134-136) and 3.1 Data base (Lines 203-206).
“This paper analyses at the end of 2018 the loan default of the portfolio of guarantees formalized for Spanish MGIs with new companies in 2003-2012, a period including economic growth end recession.”
“a long period including a cycle change with economic growth and recession, to generate financial literacy useful to promote development-oriented policies regarding SMEs.”
“The period 2003-2012 was chosen to work with practically expired portfolios, considering that the loans have an average duration of approximately 6 years. It is also a sufficiently long period to include periods of financial stress and stability.”
Reviewer 2 Report
The overall significance of the article seems to be relevant and interesting. However, I have several doubts concerning its content.
1) My general impression is that the paper is more like a research report than a scientific article. There is no research gap that could be identified in the introduction. You present a lot of managerial examples but there are no theoretical foundations where any theory would be discussed. How is the topic related to sustainability? Although there is one paragraph where you slightly explain that, I couldn't find any clear evidence.
2) There is no research goal and therefore it's difficult to understand the dominant logic of the research conducted.
3) In my opinion, the hypotheses are having no theoretical foundations. Although you mention some previous research in the background, there is totally no explanation where are your assumptions coming from. In H3 you mention "certain sectors" - which sectors? why are those chosen? that is totally not clear
4) there is no information about the data - how was it collected? were those companies family-owned? how about the selection criteria? In the title you mention new family-owned companies - how was that defined? and was your sample considering that definition?
5) you classify the purpose of the loan into two groups - how? what was the definition?
6) I'm totally surprised by your hypotheses testing - why are the H1/H2/H3 formulated in that way: "they do not have the same mean". Besides the fact that you have developed totally different hypotheses previously, such expression is not methodologically correct.
7) there is no theoretical contribution
To sum up, I believe that there is a lot of interesting analysis that you conducted but I strongly recommend rewriting the article based on a theoretical framework that is necessary for JCR articles.
Author Response
Report for Reviewer 2
The overall significance of the article seems to be relevant and interesting. However, I have several doubts concerning its content.
First of all, thank you very much for your valuable comments. All your suggestions are very important, which have important significance for our writing and scientific research work. When revising the article, we have carefully considered what you have said.
1) My general impression is that the paper is more like a research report than a scientific article. There is no research gap that could be identified in the introduction. You present a lot of managerial examples but there are no theoretical foundations where any theory would be discussed. How is the topic related to sustainability? Although there is one paragraph where you slightly explain that, I couldn't find any clear evidence.
Thank you for reading this article carefully and making valuable suggestions. As you said, the topic was not well-connected with the concept of sustainability in the original manuscript. Following your recommendations, the relationship between the manuscript and the concept of sustainability has been described in depth and supported with additional references. New paragraphs have been included in section 1 - Introduction to reinforce the conceptual framework highlighting the approach of the 17 UN GDS and to explain in more detail linking the concept of sustainability finance in the European Union context with the manuscript, including the support to MGIs Systems through the European Investment Fund (Lines 30-66; 114-117).
“Sustainable finance in the European Union is imperative to improve the contribution of finance for sustainable, inclusive growth (SDG8). The financial system has a key role to play and the EU wants the current financial system to be better aligned with its sustainable growth policies and to protect the financial system from sustainability risks [1-3].
Broadly speaking, the sustainability framework is based on environmental, economic and social issues related to present and future generations, including sustainable finance to make economic prosperity long-lasting [4-8]. Sustainable finance is strongly linked to financial sustainability, which is related to structural, relational and cognitive aspects that create value for a company [9] and the long-time survival capacity of companies. The difference between these issues is a qualitative shift that lies in implementing the sustainability of financial activities, including promoting social and environmental responsibility [10-11].
The Goal Development Sustainability (GDS) framework provides an umbrella that encompasses both ideas. Considering GDS, most social innovations case studies have been related to an improvement in health and well-being [12]. However, the importance of small and medium enterprises (SMEs) for economic development through wealth distribution, creation of employment, technological advancement, reduction of poverty and innovation [13] has contributed to the consideration of the development and sustainability in the GDS framework.
Specifically, these goals are: 8.3. to promote development-oriented policies that support productive activities, decent job creation, entrepreneurship, creativity and innovation, and encourage the formalization and growth of micro-, small- and medium-sized enterprises, including access to financial services, 8.10; to strengthen the capacity of domestic financial institutions to order to encourage and expand access to banking, insurance and financial services for all, and 9.3. to increase access for small-scale industrial and other enterprises, especially in developing countries, to financial services, including affordable credit, and their integration into value chains and markets [14].
These specific goals are to promote the development of SMEs and ensure their access to credit so that they can be sustained in the long term. In this regard, the approach of social sustainability and intergenerational transition in family businesses has been a much studied topic in the last ten years [15-17]. However, the succession of control in family businesses is especially complex [18-19], as they comprise multi-dimensional scenarios that include financial issues. Access to credit so that SMEs can survive, especially in financial stress contexts, is a serious problem, even in developed countries [20]. In this sense, one of the main problems faced by SMEs, especially young companies, is the lack of collateral, which hinders access to finance. This question is essential to ensure the sustainability of family run Small and Medium Enterprises (SMEs). On one hand, the availability of external funds is necessary for the creation and impulse of new businesses. On the other hand, credit is essential for the maintenance of these firms during their first years of life or under conditions of financial stress and to ensure intergenerational transition in these businesses [21].”
“In turn, CERSA shares the risk with The European Investment Fund (EIF) which offers securitizations of SME debt finance portfolios with guarantee institutions, this allows CERSA to take more risks with Spanish MGIs, and these MGIs can provide more loans to the SMEs [47]. CERSA has the EIF securitization of their annul portfolios since 2000 [48]”
The research gap has been exposed in section 1- Introduction (Lines 122-132).
“Despite the importance of these institutions regarding the development and financial sustainability of SMEs, MGIs have not been studied in depth. This implies the existence of a large gap in financial information regarding the default behaviour supported by these institutional guarantees. The role of the knowledge-based resources in promoting sustainability in SMEs is a very contemporary issue and financial literacy has been considered key for financial decision making [13]. In the context of MGIs, this supposes two problems: MGIs need to have enough resources available to respond to the responsibility assumed and cover the losses related to defaulting on the granted loan, but there is no information available regarding this, and (2) the lack of information regarding default makes it very difficult to promote development-oriented policies. The need for pinpoint financial knowledge regarding the financing of SMEs by MGIs in Spain was the main incentive for this research.”
We would be extremely grateful if you could give your valuable advice again.
2) There is no research goal and therefore it's difficult to understand the dominant logic of the research conducted.
Thank you for your careful reading and valuable comments. Following your valuable recommendation, a little change on the research goal has done to clarify it in the Abstract (Lines 17-19) and in the Introduction (Lines 134-136).
“This paper analyses at the end of 2018 the loan default of the portfolio of guarantees formalized for Spanish MGIs with new companies in 2003-2012, a period including economic growth end recession.”
“during 2003-2012, a long period including a cycle change with economic growth and recession, to generate financial literacy useful to promote development-oriented policies regarding SMEs.”
The logic and incentives that the study pursues have been reinforced with the paragraph related to the research gap in the Introduction (Lines 122-132), as exposed in the before response; and it has been connected to the conclusions (Lines 463-468), in order to clarify the logic of the research conducted.
“Despite the importance of these institutions regarding the development and financial sustainability of SMEs, MGIs have not been studied in depth. This implies the existence of a large gap in financial information regarding the default behaviour supported by these institutional guarantees. The role of the knowledge-based resources in promoting sustainability in SMEs is a very contemporary issue and financial literacy has been considered key for financial decision making [13]. In the context of MGIs, this supposes two problems: MGIs need to have enough resources available to respond to the responsibility assumed and cover the losses related to defaulting on the granted loan, but there is no information available regarding this, and (2) the lack of information regarding default makes it very difficult to promote development-oriented policies. The need for pinpoint financial knowledge regarding the financing of SMEs by MGIs in Spain was the main incentive for this research.”
“Thus, these results could help with the attainment of some GDS while contributing to improve access to credit for SMEs in terms of quality and cost, promoting development-oriented policies that support productive activities, decent job creation and entrepreneurship, strengthen the capacity of domestic financial institutions to encourage and expand access to banking, insurance and financial services for all and improve access to financial services, including affordable credit, for small-scale industrial and other enterprises.”
3) In my opinion, the hypotheses are having no theoretical foundations. Although you mention some previous research in the background, there is totally no explanation where are your assumptions coming from. In H3 you mention "certain sectors" - which sectors? why are those chosen? that is totally not clear.
The theoretical foundation for the hypotheses have be improved (Lines 156-167, 172-175)
“[64] concluded that larger companies show a higher index of self-finance. However, smaller companies display a precarious financial behaviour, apparently based on short-term financing. [65] found that the larger the company, the more likely it is that the company is diversified and its financial flows are less volatile. Nevertheless, [66] argued that asymmetric information problems are greater in SMEs, which leads them to endure financial constraints, which translate into a greater tendency to short-term financing and less in the long-term. These authors conclude that business size has an influence on short-term, but not on long-term debt. [67] considered that larger enterprises present higher capitalization rates and, consequently, lower levels of borrowing. This disparity in the results of empirical work in terms of size reflects a marked ambiguity in the relationship between size and indebtedness in SMEs due to the complex relationships that link them.
The influence of the sector of activity has also been widely studied, but no conclusive results have been obtained either. “
“[66] point out that sectors where assets are mostly intangible and risky tend to borrow little, as opposed to sectors where assets are tangible and relatively safe. [67] concluded that there were no significant differences between groups, except in the construction sector; moreover industrial enterprises had higher borrowing rates than service sector enterprises.”
Moreover, the hypotheses proposed in the original manuscript have been reformulated as it is explained in the response at your comment 6)
4) there is no information about the data - how was it collected? were those companies family-owned? how about the selection criteria? In the title you mention new family-owned companies - how was that defined? and was your sample considering that definition?
Data base is the population of all the new companies guaranteed by MGIs in Spain and granted by CERSA. This company assumes in Spain a percentage of loan default granted by MGIs, as it is shown in the Introduction (Line 110-111).
“In Spain, the Compañía Española de Reafianzamiento (CERSA) is the institution that assumes a percentage of loan default granted by MGIs.”
CERSA gave to the researchers the data base for this study at the end of 2018 which includes the portfolio of guarantees assessed with new companies in 2003-2012. We have explained the information collected in this data base to clarify your this question (Lines 202-213).
“CERSA defines start-ups as newly created companies with less than 3 years of activity on the date of formalization of the guaranteed loan. The period 2003-2012 was chosen to work with practically expired portfolios, considering that the loans have an average duration of approximately 6 years. It is also a sufficiently long period to include periods of financial stress and stability. The database contains cross-sectional data, where each line of information comprises a company’s financial operation and includes the characteristics of the financing guaranteed by the MGI: amount, date of formalisation and maturity, purpose of the operation (investment or working capital) and amount of the default; and the characteristics of the guaranteed company: years in business, number of employees and sector of activity. Usually, the formalised operations are associated with a single company, although there is the case of several loans from/to the same company, so the database is not of companies, but of loans guaranteed by SGR.”
The Importance of Small Family Business in Europe and the link between SMEs and family businesses is shown in the Introduction (Lines 77-87).
“The immense importance of Small Family Businesses in Europe is unquestionable [11,27,28]. In terms of relative importance, SMEs form the basis of the European economy. Family businesses make up more than 70% of all European enterprises and play an important role in the dynamism and strength of European economies, sustainability and long-term stability [29].
The link between SMEs and family businesses is evident and many of the challenges facing family firms are shared with SMEs. In Europe, family enterprises are dominated by micro enterprises and SMEs [30]. In Italy, 93% of manufacturing companies with less than 50 employees are family businesses. Family companies comprise 60% of all enterprises in Germany and provide 55% of the GDP [29]. In Spain the importance of family businesses is increasing as 95% of SMEs are family companies. These businesses contribute about 57% of the GDP and generate about 67% of private employment [30].”
5) you classify the purpose of the loan into two groups - how? what was the definition?
Thank you very much for this question. The purpose of the loan is classified into: investment capital and working capital. The purpose refers to the destination; credits for new investment are into investment capital group and credits for current assets or debt refinancing are into the working capital group. If you consider it is important to include this definition in the paper to clarify these concepts, please, let us know.
6) I'm totally surprised by your hypotheses testing - why are the H1/H2/H3 formulated in that way: "they do not have the same mean". Besides the fact that you have developed totally different hypotheses previously, such expression is not methodologically correct.
Thank you for your recommendation, the hypotheses proposed in the original manuscript have been reformulated for two new hypotheses in order to clarify the logic of the research and to improve the quality of the manuscript, following the recommendation of the reviewers. We have added a first hypothesis solved with the ratio analysis and a second hypothesis solved with the analysis of variance (Lines 185-196).
“Based on the revised background, two hypotheses are proposed in this study to identify the behaviour of defaults over time, in periods of expansion and recession, and whether the purpose of the loan, company size and economic activity, have significant effects on the volume of defaults.
H1: Default behaviour in SME credits guaranteed by MGIs changes in a different way in contexts of stability or financial stress.
H2: Default behaviour in SME credits guaranteed by MGIs is influenced significantly by the purpose of the loan, company size and business sector.
The first hypothesis was validated using a descriptive approach. The second hypothesis was validated using a stochastic analysis.
From these hypotheses, it is possible to describe the evolution of default behaviour in relation to time in the considered period, and to conclude whether the purpose of the loan, company size and/or economic activity significantly influence the defaults of family companies.”
7) there is no theoretical contribution
Thank you very much for your valuable comments. You're really considerate. Your suggestions have greatly improved our manuscript. The theoretical framework and the research gap have been reinforced (Lines 30-66, 114-117, 122-132, 134-136).
“Sustainable finance in the European Union is imperative to improve the contribution of finance for sustainable, inclusive growth (SDG8). The financial system has a key role to play and the EU wants the current financial system to be better aligned with its sustainable growth policies and to protect the financial system from sustainability risks [1-3].
Broadly speaking, the sustainability framework is based on environmental, economic and social issues related to present and future generations, including sustainable finance to make economic prosperity long-lasting [4-8]. Sustainable finance is strongly linked to financial sustainability, which is related to structural, relational and cognitive aspects that create value for a company [9] and the long-time survival capacity of companies. The difference between these issues is a qualitative shift that lies in implementing the sustainability of financial activities, including promoting social and environmental responsibility [10-11].
The Goal Development Sustainability (GDS) framework provides an umbrella that encompasses both ideas. Considering GDS, most social innovations case studies have been related to an improvement in health and well-being [12]. However, the importance of small and medium enterprises (SMEs) for economic development through wealth distribution, creation of employment, technological advancement, reduction of poverty and innovation [13] has contributed to the consideration of the development and sustainability in the GDS framework.
Specifically, these goals are: 8.3. to promote development-oriented policies that support productive activities, decent job creation, entrepreneurship, creativity and innovation, and encourage the formalization and growth of micro-, small- and medium-sized enterprises, including access to financial services, 8.10; to strengthen the capacity of domestic financial institutions to order to encourage and expand access to banking, insurance and financial services for all, and 9.3. to increase access for small-scale industrial and other enterprises, especially in developing countries, to financial services, including affordable credit, and their integration into value chains and markets [14].
These specific goals are to promote the development of SMEs and ensure their access to credit so that they can be sustained in the long term. In this regard, the approach of social sustainability and intergenerational transition in family businesses has been a much studied topic in the last ten years [15-17]. However, the succession of control in family businesses is especially complex [18-19], as they comprise multi-dimensional scenarios that include financial issues. Access to credit so that SMEs can survive, especially in financial stress contexts, is a serious problem, even in developed countries [20]. In this sense, one of the main problems faced by SMEs, especially young companies, is the lack of collateral, which hinders access to finance. This question is essential to ensure the sustainability of family run Small and Medium Enterprises (SMEs). On one hand, the availability of external funds is necessary for the creation and impulse of new businesses. On the other hand, credit is essential for the maintenance of these firms during their first years of life or under conditions of financial stress and to ensure intergenerational transition in these businesses [21].”
“In turn, CERSA shares the risk with The European Investment Fund (EIF) which offers securitizations of SME debt finance portfolios with guarantee institutions, this allows CERSA to take more risks with Spanish MGIs, and these MGIs can provide more loans to the SMEs [47]. CERSA has the EIF securitization of their annul portfolios since 2000 [48]”
“Despite the importance of these institutions regarding the development and financial sustainability of SMEs, MGIs have not been studied in depth. This implies the existence of a large gap in financial information regarding the default behaviour supported by these institutional guarantees. The role of the knowledge-based resources in promoting sustainability in SMEs is a very contemporary issue and financial literacy has been considered key for financial decision making [13]. In the context of MGIs, this supposes two problems: MGIs need to have enough resources available to respond to the responsibility assumed and cover the losses related to defaulting on the granted loan, but there is no information available regarding this, and (2) the lack of information regarding default makes it very difficult to promote development-oriented policies. The need for pinpoint financial knowledge regarding the financing of SMEs by MGIs in Spain was the main incentive for this research.”
“during 2003-2012, a long period including a cycle change with economic growth and recession, to generate financial literacy useful to promote development-oriented policies regarding SMEs.”
The social and economic implications of the study findings have been better shown in section 5 Discussion and Conclusions. In 5.1 Global analysis of loan default, it was exposed the importance of the information provided in this study for the policies aimed at financing the coverage of loan default to SMEs in four directions (Lines 387-395), one more has been added (Lines 395-396).
“and finally e) percentage of the EIF securitization to CERSA in the annual portfolio to this company.”
A new paragraph was added in Conclusions (Lines 463-468).
“Thus, these results could help with the attainment of some GDS while contributing to improve access to credit for SMEs in terms of quality and cost, promoting development-oriented policies that support productive activities, decent job creation and entrepreneurship, strengthen the capacity of domestic financial institutions to encourage and expand access to banking, insurance and financial services for all and improve access to financial services, including affordable credit, for small-scale industrial and other enterprises.”
Reviewer 3 Report
Please find my comments attached below.

Author Response
It was sent by mail
Round 2
Reviewer 1 Report
There are some major concerns regarding the following aspects:
Though the authors tried to explain this issue (sustainabilityand sustainable financeseem to be used interchangeably in this research? It is not clear at all how the ”sustainable finance”(line 14) paradigm is reflected within this research), they added more terms/notions that cannot to be found (e.g. Goal Development Sustainability – line 46) that imply a rather scarce sustainability background for the research paper. More specifically, there is no a well-defined connection between the research objectives/the research design and sustainability and sustainable development as understood from the Aims of the journal. Again, for the nowadays mainstream research ANOVA is rather basic and more complex/insightful techniques are generally employed. Of course, in relation with the hypothesis, it might be useful, but this further emphasize a rather simplistic approach that surely needs additional research insights. In terms of language and scientific presentation, there must be improvements.
Author Response
Report for Review 1
There are some major concerns regarding the following aspects:
First of all, thank you very much for your valuable comments. All your suggestions are very important and help to focus better the scientific research work. We have considered carefully what you said.
Though the authors tried to explain this issue (sustainability and sustainable finance seem to be used interchangeably in this research? It is not clear at all how the ”sustainable finance”(line 14) paradigm is reflected within this research), they added more terms/notions that cannot to be found (e.g. Goal Development Sustainability – line 46) that imply a rather scarce sustainability background for the research paper. More specifically, there is no a well-defined connection between the research objectives/the research design and sustainability and sustainable development as understood from the Aims of the journal.
We would like to thank you for your valuable contributions to improve this manuscript and to focus better keywords, the background of the research paper and its contribution to the sustainability development and the research objectives. A little change has been made in keywords (lines 26-27):
“Keywords: family new businesses; financial sustainability; access to finance, Mutual Guarantee Institutions; loan default”.
The first paragraph of the abstract has been changed (lines 12-16):
“Small family-owned companies stand the most common form of European business structure and are characterized by their long-term orientation, so they can play an important role in the new business launch related to sustainable growth. However, accessing to finance is difficult for new start-ups. Mutual Guarantee Institutions (MGIs) mitigate this problem facilitating long-term guaranteed loans to entrepreneurs, but they have to assume the default losses”.
The Introduction has been deeply reformed then we suggest it could be interesting to read it again, although the most important changes have been the first 4 paragraphs (lines 30-54):
“Sustainable finance in the European Union is imperative to improve the contribution of finance for sustainable, inclusive growth, Goal 8 at the 2030 Agenda for Sustainable Development. The EU wants the current financial system to be better aligned with its sustainable growth policies and to protect the financial system from sustainability risks [1-3] and all financial intermediary (FIs) have a key role to play to get this objective.
Sustainable finance is strongly linked to financial sustainability, which is related to structural, relational and cognitive aspects that create value for a company [4] and the long-time survival capacity of companies. The linkage between the business sustainability and the financial performance of a firm has been revealed by several studies as [5] showed.
The Sustainability Development Goals (SDG) aim development-oriented policies to foster the formalization of micro-, small- and medium-sized enterprise in SDG8. Accessing financial services and expanding access to banking are specifically included in these policies [6].
Access to credit so that SMEs can survive, especially in financial stress contexts, is a serious problem, it is more difficult and costly for SMEs to access credit than it is for larger companies, even in developed countries [7,8]. In this sense, one of the main problems faced by SMEs, especially young companies, is the lack of collateral, which hinders access to finance. This question is essential to ensure the sustainability of family run Small and Medium Enterprises (SMEs). On one hand, the availability of external funds is necessary for the creation and impulse of new businesses. On the other hand, credit is essential for the maintenance of these firms during their first years of life or under conditions of financial stress and to ensure intergenerational transition in these businesses [9]. Likewise insufficient financial literacy has been recognized as one of the main reasons for the failure of SMEs [10] and as a barrier to the sustainable development for these companies [7]. [11] identified the impact of the lack of financial on business organizations in the starting up. The role of the knowledge-based resources in promoting sustainability in SMEs is a very contemporary issue and financial literacy has been considered key for financial decision making [7].”
We have included two important aspects about the Spanish MGI that were not included in the first manuscript version for understanding better the way of operation of MGI (lines 97-100).
“The Spanish MGs are considered as FIs (supervised by Spanish central bank), provide guarantees for long-term loans and also give advice and financial literacy for the SMEs [37]. Some of them have created a department to development this area, as ELKARGI, the first one in the activity ranking [38].”
We have focused better the research gap, the objectives and the contribution of the study (lines 114-127 and lines 131-135).
“Despite the importance of these institutions regarding the development and financial sustainability of SMEs, MGIs have not been studied in depth, especially the cost of loan defaults, a key aspect for the sustainability of these FIs. To our knowledge, there is no recent research about this subject. This implies the existence of a large gap in financial information regarding the default behaviour supported by these institutional guarantees In the context of MGIs, this supposes two problems: MGIs need to have enough resources available to respond to the responsibility assumed and cover the losses related to defaulting on the granted loan, but there is no information available regarding this, and (2) the lack of information regarding default makes it very difficult to promote development-oriented policies. The need for pinpoint financial knowledge regarding the financing of SMEs by MGIs in Spain was the main incentive for this research.
The objective of this study is to characterize loan defaults and the losses related to financing provided by Spanish MGIs to new small family businesses during 2003-2012, a long period including a cycle change with economic growth and recession and to address a gap in the literature about MGIs.”
“This study contributes to the literature on access to finance and defaults costs of the new SMEs, useful to promote development-oriented policies regarding SDGs. Further the research expands knowledge about MGIs contribution to access to credit, advice and financial literacy to SMES; and the findings provide relevant information for the financial sustainability of these FIs, valuable for managers and Public Administrations, including EIF.”
Again, for the nowadays mainstream research ANOVA is rather basic and more complex/insightful techniques are generally employed. Of course, in relation with the hypothesis, it might be useful, but this further emphasize a rather simplistic approach that surely needs additional research insights. In terms of language and scientific presentation, there must be improvements.
Our database, although referring to one moment in time are cross-sectional data, where each guaranteed operation refers to a loan, with a different amount, purpose and maturity, and which do not correspond to the same company; thus, each operation is independent of the others. To contrast H2: Default behaviour in SME credits guaranteed by MGIs is influenced significantly by the purpose of the loan, company size and business sector, it´s necessary use a method to relate continuous dependent variable with categorical independent variables. In this case, a potential stochastic analysis, like a linear regression with this characteristics of variables, is ANOVA analysis.
Reviewer 2 Report
Thank you for reading my remarks and including some of them in your revised version. Still, the theoretical framework remains week. Although you have included some references to a sustainable perspective, the theoretical foundation, as well as the research gap, were not developed. You were able to present the managerial reasoning but no existing theories were discussed deeply. The research goal is still not clearly explained. In your response you confirm adding it, however, the parts of the paper that you cite don't mention the research goal:
“This paper analyses at the end of 2018 the loan default of the portfolio of guarantees formalized for Spanish MGIs with new companies in 2003-2012, a period including economic growth end recession.”
“during 2003-2012, a long period including a cycle change with economic growth and recession, to generate financial literacy useful to promote development-oriented policies regarding SMEs.”
The theoretical contribution still remains week. I would recommend discussing relevant theory(ies) in the literature review and then explain how your research contributes to those theories.
Author Response
Report for Reviewer 2
Thank you for reading my remarks and including some of them in your revised version.
First of all, thank you very much for your valuable comments. All your suggestions are very important and help to focus better the scientific research work. We have considered carefully what you said.
Still, the theoretical framework remains week. Although you have included some references to a sustainable perspective, the theoretical foundation, as well as the research gap, were not developed. You were able to present the managerial reasoning but no existing theories were discussed deeply.
We would like to thank you for your valuable contributions to improve this manuscript, we have reflected about your comments and we have focused better the theoretical foundation. It has been necessary some changes in the keywords, the background of the research paper and its contribution to the sustainability development and the research objectives. The keywords are (lines 26-27):
“Keywords: family new businesses; financial sustainability; access to finance, Mutual Guarantee Institutions; loan default”.
The first paragraph of the abstract has been changed (lines 12-16):
“Small family-owned companies stand the most common form of European business structure and are characterized by their long-term orientation, so they can play an important role in the new business launch related to sustainable growth. However, accessing to finance is difficult for new start-ups. Mutual Guarantee Institutions (MGIs) mitigate this problem facilitating long-term guaranteed loans to entrepreneurs, but they have to assume the default losses”.
The Introduction has been deeply reformed then we suggest it could be interesting to read it again, although the most important changes have been the first 4 paragraphs (lines 30-54):
“Sustainable finance in the European Union is imperative to improve the contribution of finance for sustainable, inclusive growth, Goal 8 at the 2030 Agenda for Sustainable Development. The EU wants the current financial system to be better aligned with its sustainable growth policies and to protect the financial system from sustainability risks [1-3] and all financial intermediary (FIs) have a key role to play to get this objective.
Sustainable finance is strongly linked to financial sustainability, which is related to structural, relational and cognitive aspects that create value for a company [4] and the long-time survival capacity of companies. The linkage between the business sustainability and the financial performance of a firm has been revealed by several studies as [5] showed.
The Sustainability Development Goals (SDG) aim development-oriented policies to foster the formalization of micro-, small- and medium-sized enterprise in SDG8. Accessing financial services and expanding access to banking are specifically included in these policies [6].
Access to credit so that SMEs can survive, especially in financial stress contexts, is a serious problem, it is more difficult and costly for SMEs to access credit than it is for larger companies, even in developed countries [7,8]. In this sense, one of the main problems faced by SMEs, especially young companies, is the lack of collateral, which hinders access to finance. This question is essential to ensure the sustainability of family run Small and Medium Enterprises (SMEs). On one hand, the availability of external funds is necessary for the creation and impulse of new businesses. On the other hand, credit is essential for the maintenance of these firms during their first years of life or under conditions of financial stress and to ensure intergenerational transition in these businesses [9]. Likewise insufficient financial literacy has been recognized as one of the main reasons for the failure of SMEs [10] and as a barrier to the sustainable development for these companies [7]. [11] identified the impact of the lack of financial on business organizations in the starting up. The role of the knowledge-based resources in promoting sustainability in SMEs is a very contemporary issue and financial literacy has been considered key for financial decision making [7].”
We have included two important aspects about the Spanish MGI that were not included in the first manuscript version for understanding better the way of operation of MGI (lines 97-100).
“The Spanish MGs are considered as FIs (supervised by Spanish central bank), provide guarantees for long-term loans and also give advice and financial literacy for the SMEs [37]. Some of them have created a department to development this area, as ELKARGI, the first one in the activity ranking [38].”
We have focused better the research gap (lines 114-123).
“Despite the importance of these institutions regarding the development and financial sustainability of SMEs, MGIs have not been studied in depth, especially the cost of loan defaults, a key aspect for the sustainability of these FIs. To our knowledge, there is no recent research about this subject. This implies the existence of a large gap in financial information regarding the default behaviour supported by these institutional guarantees In the context of MGIs, this supposes two problems: MGIs need to have enough resources available to respond to the responsibility assumed and cover the losses related to defaulting on the granted loan, but there is no information available regarding this, and (2) the lack of information regarding default makes it very difficult to promote development-oriented policies. The need for pinpoint financial knowledge regarding the financing of SMEs by MGIs in Spain was the main incentive for this research.”
The research goal is still not clearly explained. In your response you confirm adding it, however, the parts of the paper that you cite don't mention the research goal:
“This paper analyses at the end of 2018 the loan default of the portfolio of guarantees formalized for Spanish MGIs with new companies in 2003-2012, a period including economic growth end recession.”
“during 2003-2012, a long period including a cycle change with economic growth and recession, to generate financial literacy useful to promote development-oriented policies regarding SMEs.” The theoretical contribution still remains week.
We have focused better the research gap and the theoretical contribution of the study (lines 124-135).
“The objective of this study is to characterize loan defaults and the losses related to financing provided by Spanish MGIs to new small family businesses during 2003-2012, a long period including a cycle change with economic growth and recession and to address a gap in the literature about MGIs. To achieve this objective we developed two analyses: (i) a global analysis of the annual evolution and the average global cost of loan default using a determinist scenario and (ii) a group comparison analysis, with a classification of credit into three groups: the purpose of the loan, company size and economic activity, hence a stochastic approach.
This study contributes to the literature on access to finance and defaults costs of the new SMEs, useful to promote development-oriented policies regarding SDGs. Further the research expands knowledge about MGIs contribution to access to credit, advice and financial literacy to SMES; and the findings provide relevant information for the financial sustainability of these FIs, valuable for managers and Public Administrations, including EIF.”
Reviewer 3 Report
The author's addressed my comments.
Author Response
Dear Editor,
First of all, we would like to thank the Editor for giving us the chance to resubmit the revised manuscript MS with ID: sustainability-598188 titled: “The Role of Mutual Guarantee Institutions in the financial sustainability of New Family-Owned Small Businesses”; considering those issues raised by the referee that were not covered properly in the previous version of the manuscript.
In this revision we have addressed all the comments of the two reviewers and the editor to whom we want to thank for the opportunity given to further improve the quality of the manuscript.
In addition to the changes made based on these recommendations, we have edited parts of the manuscript to correct minor problems with the English.
Thank you and regards,
Authors
Report for the Editor
Dear Authors,
First of all I want to say that the work you have sent is interesting, so I think it should incorporate improvements so that researchers and other stakeholders who are interested in this field of research can better understand both the objectives and the research methodology. In this sense, I believe that it is appropriate that the research work achieves improvements in the following points.
Improvement points
Keywords: Sustainable finance market, ODS
Articles interesting for read.
Cheng, Y.-F.; Mutuc, E.B.; Tsai, F.-S.; Lu, K.-H.; Lin, C.-H. Social Capital and Stock Market Participation via Technologies: The Role of Households' Risk Attitude and Cognitive Ability. Sustainability 2018, 10, 1904.
Ye, J.; Kulathunga, K. How Does Financial Literacy Promote Sustainability in SMEs? A Developing Country Perspective. Sustainability 2019, 11, 2990.
-Dear editor, thank you very much for your valuable comments and suggestions. As you have suggested, we have included the suggested Keyword (Line 26). Also, the suggested references have been quoted as follows:
Line 37: Cheng, Y.-F.; Mutuc, E.B.; Tsai, F.-S.; Lu, K.-H.; Lin, C.-H. Social Capital and Stock Market Participation via Technologies: The Role of Households' Risk Attitude and Cognitive Ability. Sustainability 2018, 10, 1904.
Lines 45, 127: Ye, J.; Kulathunga, K. How Does Financial Literacy Promote Sustainability in SMEs? A Developing Country Perspective. Sustainability 2019, 11, 2990.
-Is necessary strengthen the sustainable finance paradigm, also introducing references to the 17 UN ODS. The conceptual framework of the research should also be reinforced. Reinforcing the theme of sustainability and how this article has connection with the broad concept of sustainability and therefore of the magazine.
Thank you very much for your advice. Following your recommendations, we have reinforced the conceptual framework highlighting the approach of the 17 UN SDGs (Lines 41-54). Moreover, the relationship between the manuscript and the concept of sustainability has been described in depth and supported with additional references (Lines 30-66).
“Sustainable finance in the European Union is imperative to improve the contribution of finance for sustainable, inclusive growth (SDG8). The financial system has a key role to play and the EU wants the current financial system to be better aligned with its sustainable growth policies and to protect the financial system from sustainability risks [1-3].
Broadly speaking, the sustainability framework is based on environmental, economic and social issues related to present and future generations, including sustainable finance to make economic prosperity long-lasting [4-8]. Sustainable finance is strongly linked to financial sustainability, which is related to
structural, relational and cognitive aspects that create value for a company [9] and the long-time survival capacity of companies. The difference between these issues is a qualitative shift that lies in implementing the sustainability of financial activities, including promoting social and environmental responsibility [10-11].
The Goal Development Sustainability (GDS) framework provides an umbrella that encompasses both ideas. Considering GDS, most social innovations case studies have been related to an improvement in health and well-being [12]. However, the importance of small and medium enterprises (SMEs) for economic development through wealth distribution, creation of employment, technological advancement, reduction of poverty and innovation [13] has contributed to the consideration of the development and sustainability in the GDS framework.
Specifically, these goals are: 8.3. to promote development-oriented policies that support productive activities, decent job creation, entrepreneurship, creativity and innovation, and encourage the formalization and growth of micro-, small- and medium-sized enterprises, including access to financial services, 8.10; to strengthen the capacity of domestic financial institutions to order to encourage and expand access to banking, insurance and financial services for all, and 9.3. to increase access for small-scale industrial and other enterprises, especially in developing countries, to financial services, including affordable credit, and their integration into value chains and markets [14].
These specific goals are to promote the development of SMEs and ensure their access to credit so that they can be sustained in the long term. In this regard, the approach of social sustainability and intergenerational transition in family businesses has been a much studied topic in the last ten years [15-17]. However, the succession of control in family businesses is especially complex [18-19], as they comprise multi-dimensional scenarios that include financial issues. Access to credit so that SMEs can survive, especially in financial stress contexts, is a serious problem, even in developed countries [20]. In this sense, one of the main problems faced by SMEs, especially young companies, is the lack of collateral, which hinders access to finance. This question is essential to ensure the sustainability of family run Small and Medium Enterprises (SMEs). On one hand, the availability of external funds is necessary for the creation and impulse of new businesses. On the other hand, credit is essential for the maintenance of these firms during their first years of life or under conditions of financial stress and to ensure intergenerational transition in these businesses [21].”
-In terms of methodology, the use of ANOVA should be reasoned, especially for the analysis of the data and also to value the data for the period 2003 to 2012. In this sense, the authors should refer to whether this technique has been used to carry out research of this type.
Thank you for your careful reading and valuable comments. The ANOVA was used in the analysis to complete the descriptive analysis. The ANOVA is performed for detecting the importance of the size, industry and propose of loan in fail ratio variation. This analysis explains how much of the variation of the variable of interest is explained by named variables. This method is useful in two ways, when the independent variables are categorical, and the inter-group differences are identified and hypothesis can be verified (Lines 269-273). Therefore, the authors consider that the methodology used is appropriate for the research objectives.
“ANOVA analysis is a technique equivalent to linear regression when explanatory variables are categorical, as is the case here, and explains how much of the variation of the default volume is explained by size, sector or propose of loan. This analysis has been widely used to identify differences between groups and is considered an indicator that complements descriptive analysis.”
We have added some references that only used an ANOVA or MANCOVA analysis to develop similar studies (Lines 273-276).
“Examples of studies that only used an ANOVA or MANOVA analysis to identify differences in small enterprises are those of [64, 66]. Others used a combined descriptive analysis and an ANOVA, such as [72], which used a ratio analysis and an ANOVA to identify the effect of energy suppliers on the financial structure of companies in this sector. “
With regard to the formulation of Hypothesis, the H3 should be described as "certain sectors" and the other sectors and their weight in the research should be stated.
Thank you for your recommendation, the hypotheses proposed in the original manuscript have been reformulated for two new hypotheses in order to clarify the logic of the research and to improve the quality of the manuscript, following the recommendation of the reviewers. We have added a first hypothesis solved with the ratio analysis and a second hypothesis solved with the analysis of variance (Lines 185-196).
“Based on the revised background, two hypotheses are proposed in this study to identify the behaviour of defaults over time, in periods of expansion and recession, and whether the purpose of the loan, company size and economic activity, have significant effects on the volume of defaults.
H1: Default behaviour in SME credits guaranteed by MGIs changes in a different way in contexts of stability or financial stress.
H2: Default behaviour in SME credits guaranteed by MGIs is influenced significantly by the purpose of the loan, company size and business sector.
The first hypothesis was validated using a descriptive approach. The second hypothesis was validated using a stochastic analysis.
From these hypotheses, it is possible to describe the evolution of default behaviour in relation to time in the considered period, and to conclude whether the purpose of the loan, company size and/or economic activity significantly influence the defaults of family companies.”
- Compilation of information
A technical research sheet should be included, showing the number, if they are family businesses, age of these businesses, etc. so that the reader can better understand the scope of this interesting research.
Data base is the population of all the new companies guaranteed by MGIs in Spain and granted by CERSA. This company assumes in Spain a percentage of loan default granted by MGIs, as it is shown in the Introduction (Line 110-111).
“In Spain, the Compañía Española de Reafianzamiento (CERSA) is the institution that assumes a percentage of loan default granted by MGIs.”
CERSA gave to the researchers the data base for this study at the end of 2018 which includes the portfolio of guarantees assessed with new companies in 2003-2012. We have explained the information collected in this data base to clarify your this question (Lines 202-213).
“CERSA defines start-ups as newly created companies with less than 3 years of activity on the date of formalization of the guaranteed loan. The period 2003-2012 was chosen to work with practically expired portfolios, considering that the loans have an average duration of approximately 6 years. It is also a sufficiently long period to include periods of financial stress and stability. The database contains cross-sectional data, where each line of information comprises a company’s financial operation and includes the characteristics of the financing guaranteed by the MGI: amount, date of formalisation and maturity, purpose of the operation (investment or working capital) and amount of the default; and the characteristics
of the guaranteed company: years in business, number of employees and sector of activity. Usually, the formalised operations are associated with a single company, although there is the case of several loans from/to the same company, so the database is not of companies, but of loans guaranteed by SGR.”
The Importance of Small Family Business in Europe and the link between SMEs and family businesses is shown in the Introduction (Lines 77-87).
“The immense importance of Small Family Businesses in Europe is unquestionable [11,27,28]. In terms of relative importance, SMEs form the basis of the European economy. Family businesses make up more than 70% of all European enterprises and play an important role in the dynamism and strength of European economies, sustainability and long-term stability [29].
The link between SMEs and family businesses is evident and many of the challenges facing family firms are shared with SMEs. In Europe, family enterprises are dominated by micro enterprises and SMEs [30]. In Italy, 93% of manufacturing companies with less than 50 employees are family businesses. Family companies comprise 60% of all enterprises in Germany and provide 55% of the GDP [29]. In Spain the importance of family businesses is increasing as 95% of SMEs are family companies. These businesses contribute about 57% of the GDP and generate about 67% of private employment [30].”
Please, it is necessary to be able to clarify H1/H2/H3 , tend to confuse and seem with the same meaning. In addition to the fact that you have developed totally different hypotheses previously, such an expression is not methodologically correct. It would be very convenient if the hypotheses were more differentiated from each other.
As we explained in the answer about H3, the hypotheses proposed in the original manuscript have been reformulated for two new hypotheses in order to clarify the logic of the research and to improve the quality of the manuscript, following the recommendation of the reviewers. We have added a first hypothesis solved with the ratio analysis and a second hypothesis solved with the analysis of variance (Lines 185-196).
“Based on the revised background, two hypotheses are proposed in this study to identify the behaviour of defaults over time, in periods of expansion and recession, and whether the purpose of the loan, company size and economic activity, have significant effects on the volume of defaults.
H1: Default behaviour in SME credits guaranteed by MGIs changes in a different way in contexts of stability or financial stress.
H2: Default behaviour in SME credits guaranteed by MGIs is influenced significantly by the purpose of the loan, company size and business sector.
The first hypothesis was validated using a descriptive approach. The second hypothesis was validated using a stochastic analysis.
From these hypotheses, it is possible to describe the evolution of default behaviour in relation to time in the considered period, and to conclude whether the purpose of the loan, company size and/or economic activity significantly influence the defaults of family companies.”
Thanks for your effort
Thank you very much for your time and for your valuable contributions and recommendations. Through your comments, we have identified the shortcomings of our original paper and perfected our research. We will improve the abilities of scientific research and make more achievements according to your suggestion in future work. And we sincerely hope that we can learn more from you.
Report for Review 1
Dear authors,
Although, this research could be an interesting contribution to the current knowledge regarding the SMEs (and New Family-Owned Small Businesses), there are some major concerns regarding the following aspects:
First of all, thank you very much for your valuable comments. All your suggestions are very important, which have important significance for our writing and scientific research work. When revising the article, we have considered carefully what you said.
Financial sustainability and sustainable finance seem to be used interchangeably in this research? It is not clear at all how the ”sustainable finance” (line 14) paradigm is reflected within this research. Also, there is not emphasized the connection between the theme of the research and the Aims and Scope of the Sustainability journal (specifically, the main analysis does not highlight any sustainability issue as reflected within the Aims and Scope of the journal). More specifically, how does this analysis is of interest for the wide readership of Sustainability Journal?
We would like to thank you for your valuable contributions to improve this manuscript. As you have suggested improving the quality of the manuscript, several new paragraphs have been included in section 1 - Introduction to explain in more detail linking the concept of sustainability finance in the European Union context with the manuscript, including the support to MGIs Systems through the European Investment Fund. Furthermore, the interest of this study for the readers of Sustainability Journal has been emphasized and both the terms sustainable finance and financial sustainability have been clarified. (Lines 30-66, 114-117, 122-132, 134-136).
“Sustainable finance in the European Union is imperative to improve the contribution of finance for sustainable, inclusive growth (SDG8). The financial system has a key role to play and the EU wants the current financial system to be better aligned with its sustainable growth policies and to protect the financial system from sustainability risks [1-3].
Broadly speaking, the sustainability framework is based on environmental, economic and social issues related to present and future generations, including sustainable finance to make economic prosperity long-lasting [4-8]. Sustainable finance is strongly linked to financial sustainability, which is related to structural, relational and cognitive aspects that create value for a company [9] and the long-time survival capacity of companies. The difference between these issues is a qualitative shift that lies in implementing the sustainability of financial activities, including promoting social and environmental responsibility [10-11].
The Goal Development Sustainability (GDS) framework provides an umbrella that encompasses both ideas. Considering GDS, most social innovations case studies have been related to an improvement in health and well-being [12]. However, the importance of small and medium enterprises (SMEs) for economic development through wealth distribution, creation of employment, technological advancement, reduction of poverty and innovation [13] has contributed to the consideration of the development and sustainability in the GDS framework.
Specifically, these goals are: 8.3. to promote development-oriented policies that support productive activities, decent job creation, entrepreneurship, creativity and innovation, and encourage the formalization and growth of micro-, small- and medium-sized enterprises, including access to financial services, 8.10; to strengthen the capacity of domestic financial institutions to order to encourage and expand access to
banking, insurance and financial services for all, and 9.3. to increase access for small-scale industrial and other enterprises, especially in developing countries, to financial services, including affordable credit, and their integration into value chains and markets [14].
These specific goals are to promote the development of SMEs and ensure their access to credit so that they can be sustained in the long term. In this regard, the approach of social sustainability and intergenerational transition in family businesses has been a much studied topic in the last ten years [15-17]. However, the succession of control in family businesses is especially complex [18-19], as they comprise multi-dimensional scenarios that include financial issues. Access to credit so that SMEs can survive, especially in financial stress contexts, is a serious problem, even in developed countries [20]. In this sense, one of the main problems faced by SMEs, especially young companies, is the lack of collateral, which hinders access to finance. This question is essential to ensure the sustainability of family run Small and Medium Enterprises (SMEs). On one hand, the availability of external funds is necessary for the creation and impulse of new businesses. On the other hand, credit is essential for the maintenance of these firms during their first years of life or under conditions of financial stress and to ensure intergenerational transition in these businesses [21].” “In turn, CERSA shares the risk with The European Investment Fund (EIF) which offers securitizations of SME debt finance portfolios with guarantee institutions, this allows CERSA to take more risks with Spanish MGIs, and these MGIs can provide more loans to the SMEs [47]. CERSA has the EIF securitization of their annul portfolios since 2000 [48]”
“Despite the importance of these institutions regarding the development and financial sustainability of SMEs, MGIs have not been studied in depth. This implies the existence of a large gap in financial information regarding the default behaviour supported by these institutional guarantees. The role of the knowledge-based resources in promoting sustainability in SMEs is a very contemporary issue and financial literacy has been considered key for financial decision making [13]. In the context of MGIs, this supposes two problems: MGIs need to have enough resources available to respond to the responsibility assumed and cover the losses related to defaulting on the granted loan, but there is no information available regarding this, and (2) the lack of information regarding default makes it very difficult to promote development-oriented policies. The need for pinpoint financial knowledge regarding the financing of SMEs by MGIs in Spain was the main incentive for this research.”
“during 2003-2012, a long period including a cycle change with economic growth and recession, to generate financial literacy useful to promote development-oriented policies regarding SMEs.” Furthermore, a little change on the research goal has done to clarify it in the Abstract (Line 17-19) and at the end of the Introduction (Lines 134-136).
“This paper analyses at the end of 2018 the loan default of the portfolio of guarantees formalized for Spanish MGIs with new companies in 2003-2012, a period including economic growth end recession.”
“during 2003-2012, a long period including a cycle change with economic growth and recession, to generate financial literacy useful to promote development-oriented policies regarding SMEs.”
Global analysis of loan default (line 298-324) seems to be merely a descriptive approach (and not developed with more complex technical standards, as generally required for mainstream researches). Is there any econometric background that could support its conclusions? This could inflict upon the criteria of having analyses performed with the highest technical standards. Further, are there any hypothesis formulated regarding this analysis?
Thank you for reading this article carefully and making valuable suggestions. As we said before we have include that CERSA has the EIF securitization of their portfolios then we have considered that it is necessary and important to expose as well, in 5.1 Global analysis of loan default, the utility of the research findings for de EIF and we have included this issue (Lines 395-396). “and finally e) percentage of the EIF securitization to CERSA in the annual portfolio to this company”.
As you have said, the ratio analysis is a descriptive analysis. Although, considering the reviewed literature, the analysis carried out to develop this research is suitable and complete to characterize the failed loans in companies guaranteed by MGS in Spain. Data base is the population of all the new companies guaranteed by MGI in Spain. As the data represent the population, it is not necessary to develop more complex techniques to obtain reliable results. We have explained more about the database to clarify this question (Lines 202-213).
“CERSA defines start-ups as newly created companies with less than 3 years of activity on the date of formalization of the guaranteed loan. The period 2003-2012 was chosen to work with practically expired portfolios, considering that the loans have an average duration of approximately 6 years. It is also a sufficiently long period to include periods of financial stress and stability. The database contains cross-sectional data, where each line of information comprises a company’s financial operation and includes the characteristics of the financing guaranteed by the MGI: amount, date of formalisation and maturity, purpose of the operation (investment or working capital) and amount of the default; and the characteristics of the guaranteed company: years in business, number of employees and sector of activity. Usually, the formalised operations are associated with a single company, although there is the case of several loans from/to the same company, so the database is not of companies, but of loans guaranteed by SGR.”
On the other hand, until now, loan default behaviour in this type of company has not been studied, so it’s completely unknown. We provide a determinist approach using a ratio analysis and an ANOVA analysis, which provides a stochastic approach. Results from the ANOVA have a significance of 95% and 99% (Tables 2-5). We have provided some references regarding studies that used only ratio analysis or ANOVA analysis, or ratios and ANOVA techniques, in similar research (lines 263-266, 273-276). We are so sorry that we did not explain the use of ANOVA clearly where the analysis was applied in the original manuscript. We would really appreciate it if you would forgive our carelessness. “Analysis of financial ratios has traditionally been used in financial research and when the sample is large enough, it is sufficient to characterize investment portfolios [70, 71, 59] In this sense, [59] only used a descriptive analysis to identify the effect of moral hazard on the loan characterization of Colombian SMEs using financial ratios.”
“This analysis has been widely used to identify differences between groups and companies and is considered an indicator that complements descriptive analysis. Examples of studies that only used an ANOVA or MANOVA analysis to identify differences in small enterprises are those of [64,66]. Some more recent works used a combined descriptive analysis and an ANOVA, such as [72], which used a ratio analysis and an ANOVA to identify the effect of energy suppliers on the financial structure of companies in this sector.
In this work, an ANOVA analysis was used to identify intergroup differences. To test homoscedasticity we used a Levene test and to solve the problem of heteroscedasticity, the analysis of variance was realised using a Brown-Forsythe statistic.”
Moreover, we have changed the hypotheses proposed in the original manuscript to two new hypotheses in order to clarify the logic of the research and to improve the quality of the manuscript, following the recommendation of the reviewers. We have added a first hypothesis
solved with the ratio analysis and a second hypothesis solved with the analysis of variance (Lines 185-196).
“Based on the revised background, two hypotheses are proposed in this study to identify the behaviour of defaults over time, in periods of expansion and recession, and whether the purpose of the loan, company size and economic activity, have significant effects on the volume of defaults.
H1: Default behaviour in SME credits guaranteed by MGIs changes in a different way in contexts of stability or financial stress.
H2: Default behaviour in SME credits guaranteed by MGIs is influenced significantly by the purpose of the loan, company size and business sector.
The first hypothesis was validated using a descriptive approach. The second hypothesis was validated using a stochastic analysis.
From these hypotheses, it is possible to describe the evolution of default behaviour in relation to time in the considered period, and to conclude whether the purpose of the loan, company size and/or economic activity significantly influence the defaults of family companies.”
The originality and novelty aspects are poorly mentioned, without clearly mentioning the importance for the sustainability. More specifically, there is not properly emphasized the advance in current knowledge related to sustainability and sustainable development.
Thank you for reading this article carefully and making valuable suggestions, which have greatly improved and perfected our manuscript. The originality and novelty aspects and the importance for sustainability have been remarked on in the Introduction Section (Lines 122-132).
“Despite the importance of these institutions regarding the development and financial sustainability of SMEs, MGIs have not been studied in depth. This implies the existence of a large gap in financial information regarding the default behaviour supported by these institutional guarantees. The role of the knowledge-based resources in promoting sustainability in SMEs is a very contemporary issue and financial literacy has been considered key for financial decision making [13]. In the context of MGIs, this supposes two problems: MGIs need to have enough resources available to respond to the responsibility assumed and cover the losses related to defaulting on the granted loan, but there is no information available regarding this, and (2) the lack of information regarding default makes it very difficult to promote development-oriented policies. The need for pinpoint financial knowledge regarding the financing of SMEs by MGIs in Spain was the main incentive for this research.”
The advance in current knowledge related to sustainability has been noted in the Conclusions (Lines 463-468).
“Thus, these results could help with the attainment of some GDS while contributing to improve access to credit for SMEs in terms of quality and cost, promoting development-oriented policies that support productive activities, decent job creation and entrepreneurship, strengthen the capacity of domestic financial institutions to encourage and expand access to banking, insurance and financial services for all and improve access to financial services, including affordable credit, for small-scale industrial and other enterprises.”
Why was the ANOVA employed for this analysis? There is no scientific justification for your methodological choice with academic examples (e.g. what are the advantages and limits of this method?). Was ANOVA used in similar studies (not necessarily for SMEs)? Using more complex techniques could improve the results?
Thank you for your careful reading and valuable comments. The ANOVA was used in the analysis to complete the descriptive analysis. The ANOVA is performed for detecting the importance of the size, industry and propose of loan in fail ratio variation. This analysis explains how much of the variation of the variable of interest is explained by named variables. This method is useful in two ways, when the independent variables are categorical, and the inter-group differences are identified and hypothesis can be verified (Lines 269-273). Therefore, the authors consider that the methodology used is appropriate for the research objectives.
“ANOVA analysis is a technique equivalent to linear regression when explanatory variables are categorical, as is the case here, and explains how much of the variation of the default volume is explained by size, sector or propose of loan. This analysis has been widely used to identify differences between groups and is considered an indicator that complements descriptive analysis.”
We have added some references that only used an ANOVA or MANCOVA analysis to develop similar studies (Lines 273-276).
“Examples of studies that only used an ANOVA or MANOVA analysis to identify differences in small enterprises are those of [64, 66]. Others used a combined descriptive analysis and an ANOVA, such as [72], which used a ratio analysis and an ANOVA to identify the effect of energy suppliers on the financial structure of companies in this sector. “
Why was the sample chosen only for the 2003-2012 period? In terms of language and scientific presentation, there must be improvements. For example, what does “in a specific way” explains in lines 300-301?
Thank you very much for your advice. Following your valuable recommendations, the justification of the considered period has been explained in order to improve the scientific presentation. We chose 2003-2012 for two reasons, it is long period including a cycle change with economic growth and recession and it was important to work with practically expired portfolios, considering that the loans have an average duration of approximately 6 years, so the information provided by CERSA is the situation of the annual portfolios at the end of 2018. We have justified better this in Abstract (Line 17-19), Introduction (Lines 134-136) and 3.1 Data base (Lines 203-206).
“This paper analyses at the end of 2018 the loan default of the portfolio of guarantees formalized for Spanish MGIs with new companies in 2003-2012, a period including economic growth end recession.”
“a long period including a cycle change with economic growth and recession, to generate financial literacy useful to promote development-oriented policies regarding SMEs.”
“The period 2003-2012 was chosen to work with practically expired portfolios, considering that the loans have an average duration of approximately 6 years. It is also a sufficiently long period to include periods of financial stress and stability.”
Report for Reviewer 2
The overall significance of the article seems to be relevant and interesting. However, I have several doubts concerning its content.
First of all, thank you very much for your valuable comments. All your suggestions are very important, which have important significance for our writing and scientific research work. When revising the article, we have carefully considered what you have said.
1) My general impression is that the paper is more like a research report than a scientific article. There is no research gap that could be identified in the introduction. You present a lot of managerial examples but there are no theoretical foundations where any theory would be discussed. How is the topic related to sustainability? Although there is one paragraph where you slightly explain that, I couldn't find any clear evidence.
Thank you for reading this article carefully and making valuable suggestions. As you said, the topic was not well-connected with the concept of sustainability in the original manuscript. Following your recommendations, the relationship between the manuscript and the concept of sustainability has been described in depth and supported with additional references. New paragraphs have been included in section 1 - Introduction to reinforce the conceptual framework highlighting the approach of the 17 UN GDS and to explain in more detail linking the concept of sustainability finance in the European Union context with the manuscript, including the support to MGIs Systems through the European Investment Fund (Lines 30-66; 114-117).
“Sustainable finance in the European Union is imperative to improve the contribution of finance for sustainable, inclusive growth (SDG8). The financial system has a key role to play and the EU wants the current financial system to be better aligned with its sustainable growth policies and to protect the financial system from sustainability risks [1-3].
Broadly speaking, the sustainability framework is based on environmental, economic and social issues related to present and future generations, including sustainable finance to make economic prosperity long-lasting [4-8]. Sustainable finance is strongly linked to financial sustainability, which is related to structural, relational and cognitive aspects that create value for a company [9] and the long-time survival capacity of companies. The difference between these issues is a qualitative shift that lies in implementing the sustainability of financial activities, including promoting social and environmental responsibility [10-11].
The Goal Development Sustainability (GDS) framework provides an umbrella that encompasses both ideas. Considering GDS, most social innovations case studies have been related to an improvement in health and well-being [12]. However, the importance of small and medium enterprises (SMEs) for economic development through wealth distribution, creation of employment, technological advancement, reduction of poverty and innovation [13] has contributed to the consideration of the development and sustainability in the GDS framework.
Specifically, these goals are: 8.3. to promote development-oriented policies that support productive activities, decent job creation, entrepreneurship, creativity and innovation, and encourage the formalization and growth of micro-, small- and medium-sized enterprises, including access to financial services, 8.10; to strengthen the capacity of domestic financial institutions to order to encourage and expand access to banking, insurance and financial services for all, and 9.3. to increase access for small-scale industrial and other enterprises, especially in developing countries, to financial services, including affordable credit, and their integration into value chains and markets [14].
These specific goals are to promote the development of SMEs and ensure their access to credit so that they can be sustained in the long term. In this regard, the approach of social sustainability and intergenerational transition in family businesses has been a much studied topic in the last ten years [15-17]. However, the succession of control in family businesses is especially complex [18-19], as they comprise multi-dimensional scenarios that include financial issues. Access to credit so that SMEs can survive, especially in financial stress contexts, is a serious problem, even in developed countries [20]. In this sense, one of the main problems faced by SMEs, especially young companies, is the lack of collateral, which hinders access to finance. This question is essential to ensure the sustainability of family run Small and Medium Enterprises (SMEs). On one hand, the availability of external funds is necessary for the creation and impulse of new businesses. On the other hand, credit is essential for the maintenance of these firms during their first years of life or under conditions of financial stress and to ensure intergenerational transition in these businesses [21].”
“In turn, CERSA shares the risk with The European Investment Fund (EIF) which offers securitizations of SME debt finance portfolios with guarantee institutions, this allows CERSA to take more risks with Spanish MGIs, and these MGIs can provide more loans to the SMEs [47]. CERSA has the EIF securitization of their annul portfolios since 2000 [48]”
The research gap has been exposed in section 1- Introduction (Lines 122-132).
“Despite the importance of these institutions regarding the development and financial sustainability of SMEs, MGIs have not been studied in depth. This implies the existence of a large gap in financial information regarding the default behaviour supported by these institutional guarantees. The role of the knowledge-based resources in promoting sustainability in SMEs is a very contemporary issue and financial literacy has been considered key for financial decision making [13]. In the context of MGIs, this supposes two problems: MGIs need to have enough resources available to respond to the responsibility assumed and cover the losses related to defaulting on the granted loan, but there is no information available regarding this, and (2) the lack of information regarding default makes it very difficult to promote development-oriented policies. The need for pinpoint financial knowledge regarding the financing of SMEs by MGIs in Spain was the main incentive for this research.”
We would be extremely grateful if you could give your valuable advice again.
2) There is no research goal and therefore it's difficult to understand the dominant logic of the research conducted.
Thank you for your careful reading and valuable comments. Following your valuable recommendation, a little change on the research goal has done to clarify it in the Abstract (Lines 17-19) and in the Introduction (Lines 134-136).
“This paper analyses at the end of 2018 the loan default of the portfolio of guarantees formalized for Spanish MGIs with new companies in 2003-2012, a period including economic growth end recession.”
“during 2003-2012, a long period including a cycle change with economic growth and recession, to generate financial literacy useful to promote development-oriented policies regarding SMEs.”
The logic and incentives that the study pursues have been reinforced with the paragraph related to the research gap in the Introduction (Lines 122-132), as exposed in the before response; and it has been connected to the conclusions (Lines 463-468), in order to clarify the logic of the research conducted.
“Despite the importance of these institutions regarding the development and financial sustainability of SMEs, MGIs have not been studied in depth. This implies the existence of a large gap in financial information regarding the default behaviour supported by these institutional guarantees. The role of the knowledge-based resources in promoting sustainability in SMEs is a very contemporary issue and financial literacy has been considered key for financial decision making [13]. In the context of MGIs, this supposes two problems: MGIs need to have enough resources available to respond to the responsibility assumed and cover the losses related to defaulting on the granted loan, but there is no information available regarding this, and (2) the lack of information regarding default makes it very difficult to promote development-oriented policies. The need for pinpoint financial knowledge regarding the financing of SMEs by MGIs in Spain was the main incentive for this research.”
“Thus, these results could help with the attainment of some GDS while contributing to improve access to credit for SMEs in terms of quality and cost, promoting development-oriented policies that support
productive activities, decent job creation and entrepreneurship, strengthen the capacity of domestic financial institutions to encourage and expand access to banking, insurance and financial services for all and improve access to financial services, including affordable credit, for small-scale industrial and other enterprises.”
3) In my opinion, the hypotheses are having no theoretical foundations. Although you mention some previous research in the background, there is totally no explanation where are your assumptions coming from. In H3 you mention "certain sectors" - which sectors? why are those chosen? that is totally not clear.
The theoretical foundation for the hypotheses have be improved (Lines 156-167, 172-175)
“[64] concluded that larger companies show a higher index of self-finance. However, smaller companies display a precarious financial behaviour, apparently based on short-term financing. [65] found that the larger the company, the more likely it is that the company is diversified and its financial flows are less volatile. Nevertheless, [66] argued that asymmetric information problems are greater in SMEs, which leads them to endure financial constraints, which translate into a greater tendency to short-term financing and less in the long-term. These authors conclude that business size has an influence on short-term, but not on long-term debt. [67] considered that larger enterprises present higher capitalization rates and, consequently, lower levels of borrowing. This disparity in the results of empirical work in terms of size reflects a marked ambiguity in the relationship between size and indebtedness in SMEs due to the complex relationships that link them.
The influence of the sector of activity has also been widely studied, but no conclusive results have been obtained either. “
“[66] point out that sectors where assets are mostly intangible and risky tend to borrow little, as opposed to sectors where assets are tangible and relatively safe. [67] concluded that there were no significant differences between groups, except in the construction sector; moreover industrial enterprises had higher borrowing rates than service sector enterprises.”
Moreover, the hypotheses proposed in the original manuscript have been reformulated as it is explained in the response at your comment 6)
4) there is no information about the data - how was it collected? were those companies family-owned? how about the selection criteria? In the title you mention new family-owned companies - how was that defined? and was your sample considering that definition?
Data base is the population of all the new companies guaranteed by MGIs in Spain and granted by CERSA. This company assumes in Spain a percentage of loan default granted by MGIs, as it is shown in the Introduction (Line 110-111).
“In Spain, the Compañía Española de Reafianzamiento (CERSA) is the institution that assumes a percentage of loan default granted by MGIs.”
CERSA gave to the researchers the data base for this study at the end of 2018 which includes the portfolio of guarantees assessed with new companies in 2003-2012. We have explained the information collected in this data base to clarify your this question (Lines 202-213).
“CERSA defines start-ups as newly created companies with less than 3 years of activity on the date of formalization of the guaranteed loan. The period 2003-2012 was chosen to work with practically expired portfolios, considering that the loans have an average duration of approximately 6 years. It is also a sufficiently long period to include periods of financial stress and stability. The database contains cross-
sectional data, where each line of information comprises a company’s financial operation and includes the characteristics of the financing guaranteed by the MGI: amount, date of formalisation and maturity, purpose of the operation (investment or working capital) and amount of the default; and the characteristics of the guaranteed company: years in business, number of employees and sector of activity. Usually, the formalised operations are associated with a single company, although there is the case of several loans from/to the same company, so the database is not of companies, but of loans guaranteed by SGR.”
The Importance of Small Family Business in Europe and the link between SMEs and family businesses is shown in the Introduction (Lines 77-87).
“The immense importance of Small Family Businesses in Europe is unquestionable [11,27,28]. In terms of relative importance, SMEs form the basis of the European economy. Family businesses make up more than 70% of all European enterprises and play an important role in the dynamism and strength of European economies, sustainability and long-term stability [29].
The link between SMEs and family businesses is evident and many of the challenges facing family firms are shared with SMEs. In Europe, family enterprises are dominated by micro enterprises and SMEs [30]. In Italy, 93% of manufacturing companies with less than 50 employees are family businesses. Family companies comprise 60% of all enterprises in Germany and provide 55% of the GDP [29]. In Spain the importance of family businesses is increasing as 95% of SMEs are family companies. These businesses contribute about 57% of the GDP and generate about 67% of private employment [30].”
5) you classify the purpose of the loan into two groups - how? what was the definition?
Thank you very much for this question. The purpose of the loan is classified into: investment capital and working capital. The purpose refers to the destination; credits for new investment are into investment capital group and credits for current assets or debt refinancing are into the working capital group. If you consider it is important to include this definition in the paper to clarify these concepts, please, let us know.
6) I'm totally surprised by your hypotheses testing - why are the H1/H2/H3 formulated in that way: "they do not have the same mean". Besides the fact that you have developed totally different hypotheses previously, such expression is not methodologically correct.
Thank you for your recommendation, the hypotheses proposed in the original manuscript have been reformulated for two new hypotheses in order to clarify the logic of the research and to improve the quality of the manuscript, following the recommendation of the reviewers. We have added a first hypothesis solved with the ratio analysis and a second hypothesis solved with the analysis of variance (Lines 185-196).
“Based on the revised background, two hypotheses are proposed in this study to identify the behaviour of defaults over time, in periods of expansion and recession, and whether the purpose of the loan, company size and economic activity, have significant effects on the volume of defaults.
H1: Default behaviour in SME credits guaranteed by MGIs changes in a different way in contexts of stability or financial stress.
H2: Default behaviour in SME credits guaranteed by MGIs is influenced significantly by the purpose of the loan, company size and business sector.
The first hypothesis was validated using a descriptive approach. The second hypothesis was validated using a stochastic analysis.
From these hypotheses, it is possible to describe the evolution of default behaviour in relation to time in the considered period, and to conclude whether the purpose of the loan, company size and/or economic activity significantly influence the defaults of family companies.”
7) there is no theoretical contribution
Thank you very much for your valuable comments. You're really considerate. Your suggestions have greatly improved our manuscript. The theoretical framework and the research gap have been reinforced (Lines 30-66, 114-117, 122-132, 134-136).
“Sustainable finance in the European Union is imperative to improve the contribution of finance for sustainable, inclusive growth (SDG8). The financial system has a key role to play and the EU wants the current financial system to be better aligned with its sustainable growth policies and to protect the financial system from sustainability risks [1-3].
Broadly speaking, the sustainability framework is based on environmental, economic and social issues related to present and future generations, including sustainable finance to make economic prosperity long-lasting [4-8]. Sustainable finance is strongly linked to financial sustainability, which is related to structural, relational and cognitive aspects that create value for a company [9] and the long-time survival capacity of companies. The difference between these issues is a qualitative shift that lies in implementing the sustainability of financial activities, including promoting social and environmental responsibility [10-11].
The Goal Development Sustainability (GDS) framework provides an umbrella that encompasses both ideas. Considering GDS, most social innovations case studies have been related to an improvement in health and well-being [12]. However, the importance of small and medium enterprises (SMEs) for economic development through wealth distribution, creation of employment, technological advancement, reduction of poverty and innovation [13] has contributed to the consideration of the development and sustainability in the GDS framework.
Specifically, these goals are: 8.3. to promote development-oriented policies that support productive activities, decent job creation, entrepreneurship, creativity and innovation, and encourage the formalization and growth of micro-, small- and medium-sized enterprises, including access to financial services, 8.10; to strengthen the capacity of domestic financial institutions to order to encourage and expand access to banking, insurance and financial services for all, and 9.3. to increase access for small-scale industrial and other enterprises, especially in developing countries, to financial services, including affordable credit, and their integration into value chains and markets [14].
These specific goals are to promote the development of SMEs and ensure their access to credit so that they can be sustained in the long term. In this regard, the approach of social sustainability and intergenerational transition in family businesses has been a much studied topic in the last ten years [15-17]. However, the succession of control in family businesses is especially complex [18-19], as they comprise multi-dimensional scenarios that include financial issues. Access to credit so that SMEs can survive, especially in financial stress contexts, is a serious problem, even in developed countries [20]. In this sense, one of the main problems faced by SMEs, especially young companies, is the lack of collateral, which hinders access to finance. This question is essential to ensure the sustainability of family run Small and Medium Enterprises (SMEs). On one hand, the availability of external funds is necessary for the creation and impulse of new businesses. On the other hand, credit is essential for the maintenance of these firms during their first years of life or under conditions of financial stress and to ensure intergenerational transition in these businesses [21].” “In turn, CERSA shares the risk with The European Investment Fund (EIF) which offers securitizations of SME debt finance portfolios with guarantee institutions, this allows CERSA to take more risks with Spanish MGIs, and these MGIs can provide more loans to the SMEs [47]. CERSA has the EIF securitization of their annul portfolios since 2000 [48]”
“Despite the importance of these institutions regarding the development and financial sustainability of SMEs, MGIs have not been studied in depth. This implies the existence of a large gap in financial information regarding the default behaviour supported by these institutional guarantees. The role of the knowledge-based resources in promoting sustainability in SMEs is a very contemporary issue and financial literacy has been considered key for financial decision making [13]. In the context of MGIs, this supposes two problems: MGIs need to have enough resources available to respond to the responsibility assumed and cover the losses related to defaulting on the granted loan, but there is no information available regarding this, and (2) the lack of information regarding default makes it very difficult to promote development-oriented policies. The need for pinpoint financial knowledge regarding the financing of SMEs by MGIs in Spain was the main incentive for this research.”
“during 2003-2012, a long period including a cycle change with economic growth and recession, to generate financial literacy useful to promote development-oriented policies regarding SMEs.”
The social and economic implications of the study findings have been better shown in section 5 Discussion and Conclusions. In 5.1 Global analysis of loan default, it was exposed the importance of the information provided in this study for the policies aimed at financing the coverage of loan default to SMEs in four directions (Lines 387-395), one more has been added (Lines 395-396). “and finally e) percentage of the EIF securitization to CERSA in the annual portfolio to this company.”
A new paragraph was added in Conclusions (Lines 463-468).
“Thus, these results could help with the attainment of some GDS while contributing to improve access to credit for SMEs in terms of quality and cost, promoting development-oriented policies that support productive activities, decent job creation and entrepreneurship, strengthen the capacity of domestic financial institutions to encourage and expand access to banking, insurance and financial services for all and improve access to financial services, including affordable credit, for small-scale industrial and other enterprises.”
Report for Reviewer 3
1. Authors should give sample sources in the article.
Data base is the population of all the new companies guaranteed by MGIs in Spain and granted by CERSA. This company assumes in Spain a percentage of loan default granted by MGIs, as it is shown in the Introduction (Line 110-111).
“In Spain, the Compañía Española de Reafianzamiento (CERSA) is the institution that assumes a percentage of loan default granted by MGIs.”
CERSA gave to the researchers the data base for this study at the end of 2018 which includes the portfolio of guarantees assessed with new companies in 2003-2012. We have explained the information collected in this data base to clarify your this question (Lines 202-213).
“CERSA defines start-ups as newly created companies with less than 3 years of activity on the date of formalization of the guaranteed loan. The period 2003-2012 was chosen to work with practically expired portfolios, considering that the loans have an average duration of approximately 6 years. It is also a sufficiently long period to include periods of financial stress and stability. The database contains cross-sectional data, where each line of information comprises a company’s financial operation and includes the characteristics of the financing guaranteed by the MGI: amount, date of formalisation and maturity,
purpose of the operation (investment or working capital) and amount of the default; and the characteristics of the guaranteed company: years in business, number of employees and sector of activity. Usually, the formalised operations are associated with a single company, although there is the case of several loans from/to the same company, so the database is not of companies, but of loans guaranteed by SGR.”
2. The paper needs to justify theoretically for variables chosen.
The theoretical foundation for the hypotheses have be improved (Lines 156-167, 172-175)
“[64] concluded that larger companies show a higher index of self-finance. However, smaller companies display a precarious financial behaviour, apparently based on short-term financing. [65] found that the larger the company, the more likely it is that the company is diversified and its financial flows are less volatile. Nevertheless, [66] argued that asymmetric information problems are greater in SMEs, which leads them to endure financial constraints, which translate into a greater tendency to short-term financing and less in the long-term. These authors conclude that business size has an influence on short-term, but not on long-term debt. [67] considered that larger enterprises present higher capitalization rates and, consequently, lower levels of borrowing. This disparity in the results of empirical work in terms of size reflects a marked ambiguity in the relationship between size and indebtedness in SMEs due to the complex relationships that link them.
The influence of the sector of activity has also been widely studied, but no conclusive results have been obtained either. “
“[66] point out that sectors where assets are mostly intangible and risky tend to borrow little, as opposed to sectors where assets are tangible and relatively safe. [67] concluded that there were no significant differences between groups, except in the construction sector; moreover industrial enterprises had higher borrowing rates than service sector enterprises.”
Moreover, the hypotheses proposed in the original manuscript have been reformulated as it is explained in the response at your comment 6)
3. Use of deterministic analyses is a basic and just using ANOVA is not enough. The article needs to use some other stochastic methods like predictive analyses: regression or time series analyses are required for robust results. Since the data obtained from 2003-2012 period, so I suggest the use of either time series or panel estimation.
Thank you for your careful reading and valuable comments. The ANOVA was used in the analysis to complete the descriptive analysis. The ANOVA is performed for detecting the importance of the size, industry and propose of loan in fail ratio variation. This analysis explains how much of the variation of the variable of interest is explained by named variables. This method is useful in two ways, when the independent variables are categorical, and the inter-group differences are identified and hypothesis can be verified (Lines 269-273). Therefore, the authors consider that the methodology used is appropriate for the research objectives.
“ANOVA analysis is a technique equivalent to linear regression when explanatory variables are categorical, as is the case here, and explains how much of the variation of the default volume is explained by size, sector or propose of loan. This analysis has been widely used to identify differences between groups and is considered an indicator that complements descriptive analysis.”
We have added some references that only used an ANOVA or MANCOVA analysis to develop similar studies (Lines 273-276).
“Examples of studies that only used an ANOVA or MANOVA analysis to identify differences in small enterprises are those of [64, 66]. Others used a combined descriptive analysis and an ANOVA, such as [72], which used a ratio analysis and an ANOVA to identify the effect of energy suppliers on the financial structure of companies in this sector. “
Time series analyses and panel estimation are two extremely interesting forms of analysis. However, for the objective of this research they are not the most suitable. In addition, the data, although referring to one moment in time are cross-sectional, where each guaranteed operation refers to a loan, with a different amount, purpose and maturity, and which do not correspond to the same company. Each operation is independent of the others, and therefore, the analyses you propose would not be adequate for the data that were used in this investigation. However, we are very grateful for your suggestions and will take them into account for future research in which we may use other types of data.
Round 3
Reviewer 1 Report
Extensive editing is required (e.g. line 328).
Extensive editing of English language and style required.
Author Response
Dear reviewer, thank you very much for your appreciated advises. The changes were made based on your recommendation. We have edited parts of the manuscript to correct minor problems with the English.
Kind regards,
The authors.
Reviewer 2 Report
Thank you for considering all remarks, I believe that the improved version of your manuscript would be cited.
Author Response
Dear reviewer, thank you very much for your appreciatted advises. We attach here the final version of the manuscript with the changes requested.
Kind regards,
The authors
